# Not so terrifying after all? A set of failed replications of the mortality salience effects of Terror Management Theory

Stanislav Treger[1☯], Erik M. Benau[2☯]*, C. Alix Timko[3,4]

**1** Northern Trust Corporation, Chicago, IL United States of America, **2** Department of Psychology, SUNY Old Westbury, Old Westbury, NY, United States of America, **3** Department of Psychiatry, Perelman School of Medicine, University of Pennsylvania, Philadelphia, PA, United States of America, **4** Department of Psychology, Towson University, Towson, MD, United States of America

☯ These authors contributed equally to this work.
* benaue@oldwestbury.edu

**Data Availability Statement:** All of the data, the code used to prepare and to analyze the data, as well as all of the materials used in each study, are openly available at the study's OSF site at: https://

## Abstract

Terror Management Theory (TMT) postulates that humans, in response to awareness of their death, developed complex defenses to remove the salience and discomfort stemming from those thoughts. In a standard paradigm to test this theory, an individual is presented with a death-related prime (Mortality Salience; MS), such as writing the details of their own death, or something neutral, such as watching television. After a distractor task (for delay), participants complete the dependent variable, such as rating how much they like or agree with a pro- or anti-national essay and its author. Individuals in the MS condition typically exhibit greater worldview defense than control conditions by rating the pro-national essay more positively and the anti-national essay more negatively. We completed five separate studies across five unique samples with the goal of replicating and extending this well-established pattern to provide further understanding of the phenomena that underlie the effects of MS. However, despite using standard procedures, we were unable to replicate basic patterns of the dependent variable in the MS conditions. We also pooled all responses into two meta-analyses, one examining all dependent variables and one focusing on the anti-national essay; yet the effect sizes in these analyses did not significantly differ from zero. We discuss the methodological and theoretical implications of these (unintended) failures to replicate. It is not clear if these null findings were due to methodological limitations, restraints of online/crowd-sourced recruitment, or ever-evolving sociocultural factors.

## Introduction

Terror Management Theory (TMT) suggests that humans have developed diverse and complex means to reduce or prevent fear and anxieties regarding death [1, 2]. One of TMT's central premises is that aspects of societies' culture, worldviews, and bases for self-concept evolved to protect one from thoughts of death in daily life [3–6]. When a person is made aware of the inevitability of death (i.e., "Mortality Salience" or MS), or thinks about death in general (i.e.,

osf.io/QDA5B/ (DOI: 10.17605/OSF.IO/QDA5B). All stimuli are publicly available, described in the manuscript, or presented in the same repository.

**Funding:** The author(s) received no specific funding for this wor

**Competing interests:** The authors have declared that no competing interests exist

"Death Though Accessibility"), the "overwhelming terror" and anxiety that ensues can result in shifts in attitudes, beliefs, or behaviors intended to suppress or escape the terror until either the terror is reduced or one's mortality becomes less salient [7, 8].

The transient shifts in behaviors and attitudes occurring in response to MS can be drastic, even violent, and typically serve to support an in-group, punish an out-group, reinforce a positive self-concept (i.e., self-esteem), and/or defend one's worldview [9]. The purposes of these behaviors and attitude shifts is to internally reinforce a belief that one's life has meaning and that one will be remembered after death [1]. Nearly 40 years of experimental and observational data provide compelling and robust empirical support that MS can measurably change behavior, supporting the hypothesis that sociocultural and psychosocial structures may have developed with the primary function of blunting fears of death and dying [1, 10]. In short, TMT has grown into an important component of psychological study that is thought to explain broad swaths of the human experience [8, 11].

TMT suggests that three central constructs serve to reduce and cope with existential anxiety: *culture/worldview*, *close relationships*, and *self-esteem* [12, 13]. Culture/worldview and self-esteem are partially achieved through a feeling of *belonging* and inclusion. *Belonging* occurs when one successfully contributes to a culture or society, and meets and perpetuates its ideals, values, and standards [12, 13]. Self-esteem (i.e., a [positive] evaluation of one's value), in the TMT model serves to provide feedback about whether an individual contributes to these standards and, therefore, has successfully attained interpersonal belonging [4]. Earning a sense of belonging and self-esteem results in feelings of safety and meaning [13], that, in turn, result in "symbolic immortality," or the idea that an individual can figuratively "live forever" [14]. Thus, the terror resulting from thoughts of death is mitigated by one's personal success, value, and meaning achieved by being part of something bigger than oneself [15, 16]. Supporting this assertion are the results of several studies where people with high (vs. low) self-esteem and/or an extant sense of "belonging" are largely resilient to MS inductions [7, 17] and are more likely to endorse beliefs in an afterlife and their own symbolic immortality [16]. Intriguingly, religious and atheist participants have shown similar (but usually attenuated) patterns of results [14], suggesting that "symbolic immortality" is not religious, *per se*, and religious adherence may be just one means of attaining symbolic immortality [18].

## Making mortality salient

Data supporting TMT are typically obtained in studies where MS is induced, followed by a delay with a distraction task, and then the dependent variable of interest. In experimental paradigms, the most common MS induction method is with a writing prime [19, 20] wherein an individual writes two essays. In the MS condition, the participant briefly contemplates and then writes about (a) thoughts and emotions related their own death and (b) what happens as one physically dies. Control conditions in prior research have involved neutral topics such as watching TV, unpleasant experiences unrelated to death such as dental pain to rule out discomfort as a variable [20, 21], or other variables of interest, such as uncertainty [e.g., 22].

A seemingly necessary component of successful death-thought accessibility is a delay between the experimental induction (i.e., the MS task) and the measure of interest [21, 23, 24]. Numerous studies suggest that immediately following the MS induction, thoughts of death are reflexively and unconsciously suppressed (proximal effects), which, after 2–6 minutes, are ineffective due to a depletion of cognitive resources necessary to suppress these thoughts [24]. Notably, proximal defenses tend to be quite effective and longer delays result in greater defensive reactions and anxiety [21, 25, 26]. Following the delay, the inability to suppress death thoughts results in behavioral and attitudinal shifts (distal effects) that are of interest to the

researcher. To engender distal effects of death-thought accessibility, participants engage in a neutral task between the MS prime and the dependent measure, such as completing unrelated questionnaires or reading a neutral short story and answering questions [21, 24, 27].

## Individual differences in the effects of MS

Effect sizes across studies appear to vary widely across groups. In a now-canonical meta-analysis of 277 experiments, Burke, Martens (19) found that the average effect size across studies was moderate ($r = .35$), and several moderators explained variation in effect sizes. The strongest MS effects were observed in samples of Americans ($r = .37$) and college students ($r = .36$). Age did not moderate the effects despite 68% of participants having been between 17–27 years old. When using age as a moderator, the impacts of MS attenuated and even reversed (e.g., greater levels of egalitarian attitudes emerged) in samples of older adults [e.g., 28, 29]. In a more recent meta-analysis, Schindler et al. [30] identified a smaller, but significant overall pooled Hedge's $g$ of 0.34, or approximately $r = .168$. Schindler and colleagues focused on methods more than individual differences, the latter of which are discussed further below. Nevertheless, these two studies, among others, highlight substantial variation in the impact of MS on worldview defense.

Cross-cultural work has also suggested that the impact of MS may not be as universal as originally believed. Several groups who recruited primarily East Asian samples were unable to find significant effects or identified very small effects [e.g., 31–34]. These authors suggested that East Asian individuals may be more tolerant and/or fear death less than Westerners, both of which mitigate the "terror" that underlies MS inductions, potentially due to reduced individualism within these cultures. Yen and Cheng [35] reanalyzed the studies that Burke and colleagues [19] included in their meta-analysis and found that Asian samples exhibited significant effects of MS, although their pooled effect size ($r = .26$) was significantly smaller than the US ($r = .41$) or European/Israeli samples ($r = .30$). However, cultural orientation and relational (vs. personal) self-esteem have emerged as important moderators of MS effects [34], suggesting that "Western" vs. "non-Western" may not be sufficient to understand cultural influences in TMT; and classifying studies that way may be problematic and essentialist. Although infrequently studied, comparisons of samples from cultures with Judeo-Christian majorities to other religions (e.g., Muslim, Hindu) indicate that the impact of MS in other cultures may be reduced, but still present more often than not [36]. Together, these findings have led some to argue that the impact of MS is best observed in undergraduates from "Western, Education, Industrialized, Rich, and Democratic" (WEIRD) cultures [37].

## Critiques and alternative theoretical perspectives

As with nearly any scientific hypothesis or theory, TMT is not without its skeptics, critics, and dissenters [see for review: 9, 38]. Some argue it is a culturally-laden, if not culture-bound, phenomenon [e.g., 35, 37], others have challenged TMT's definition of self-esteem [39, 40], its interpretation of evolutionary principles [e.g., 41–43], its rationale for using a delay between MS and the outcome of interest [21, 24], its postulation that MS itself is unique in generating effects [44–46], and the proposed mechanisms for why MS leads to its effects [22]. There have also long been failures to replicate some of the basic effects seen in TMT literature [e.g., Ochsman, personal communication, cited in 27], even with meticulous methods, faithful experimental conditions, and enviable sample sizes [31, 32, 47–50]. Although meta-analyses indicate a fairly robust effect of TMT on a host of dependent variables [19], Schindler and colleagues [30] found that the prototypical dependent variable—worldview defense—is *not* robustly found across studies when accounting for publication bias, control conditions, researcher effects (namely,

researcher degrees of freedom), and other key components [30]. In other words, these "conceptual replications," among others, were inconsistently successful, especially in more recent studies [30, 50]. Others have accused previous researchers of cherry-picking data, presenting false-positives, and otherwise engaging in misleading or less-than-honest behavior (intentionally or otherwise) in studies of TMT [30, 49–51]. However, lapses in rigor and difficulty with replication are far from unique to TMT-related work [52, 53]. Some argue these null and/or questionable results are grounds to dismiss the theory as a false-positive [e.g., 49], yet the body of supporting evidence of TMT and its tenets precludes such a blanket dismissal.

Sustainable theories are necessarily falsifiable, and failures to replicate serve to understand the limits of any explanatory hypothesis: research within TMT is no different [1]. For example, an early replication failure noted by Ochsman and colleagues [cited in 27] resulted in a refining of methods wherein MS tasks that are intense, blatant, or rational can impede the desired effects compared to experiential and/or subtle primes [54, 55]. Later studies found that more intense primes require more time for worldview defenses to "kick-in," making subtle primes a more cost effective method than intense primes [23]. Thus, it would not be fair to say that all death primes are comparable and attention to task design and demand is necessary to better understand whether, and how, awareness of mortality influences the human experience. With these controversies and aberrant findings in mind, it is ever more important that researchers consider the reliability and validity of the instruments and techniques they use to test hypotheses derived from TMT and ensure that we have tested what we intended to test.

## The purpose of this manuscript

On the surface, the large literature on TMT indicates that the effects of MS are robust and replicable, even in small samples. Still, there may be some cause for concern given the variability in effect size as a function of sample composition and precise methodology used [19, 21, 23, 35, 56]. Here, we present a series of independent studies from multiple labs with diverse samples that all exhibited a consistent result: a failure to replicate established effects using the gold-standard MS essay prime. Although we did not originally aim to run these studies as tests of replicability, and therefore did not pre-register them as others have [50], we believe that the consistent pattern of failed replications of MS effects warrants report. It is important to note that we do not aim to challenge TMT as a theory, we are not promoting other theoretical explanations over TMT for the phenomena discussed above, nor do we propose a novel theoretical foundation to explain our findings. TMT has several strengths and limitations that may explain our (unintentional) failures to replicate these basic effects. The purpose of reporting these studies is to further our understanding of TMT, its methodology in the context of contemporary psychological science, and the conclusions drawn from within its framework.

It has long been noted that statistical significance is not, nor should be, the sole index of "successful" or worthy scientific endeavors: reporting only significant findings and failing to report non-significant findings can result in misleading conclusions and problematic follow-ups [57, 58]. In addressing criticism and critique of TMT, Pyszczynski and colleagues [1] correctly assert that scientific progress is predicated on presenting and addressing "anomalous findings" to refine a hypothesis or theory. Again, our goal is to highlight areas the field would do well to examine closer and provide additional critical inquiry.

## Methods

### Overview of studies

The institutional review boards of each university (Studies 1 and 2: DePaul University; Studies 3 and 4: Syracuse University; Study 5: Towson University) approved the methods of each

study described below. All participants completed informed consent procedures before participating. All included participants identified as United States residents, and over 98% of each sample were US citizens. Additionally, each of these studies had manipulations beyond an MS prime vs. control and evaluation of pro- or anti-American essays. For this paper, we focus on the results of the MS prime except as noted. The decision to focus on these primes was twofold: first, for brevity and second, since each study was intended to be a replication and *extension* of classic TMT work, the failure to replicate precluded continuing to the extension these researchers had hoped to investigate [59]. Additionally, all participants included in the present analyses provided complete data, including responses on the filler task(s) used to cause the requisite delay and distraction between prime and dependent variables [23, 24].

## Independent variables

Each study utilized a writing prime as the independent variable. Participants began by completing a set of neutral individual difference and demographic measures (discussed in more detail for each study), and then engaged in a writing prime standard to TMT work [e.g., 5, 21]:

1. "Please briefly describe the emotions that the thought of [MS/control condition] arouses in you;"

2. "Write down, as specifically as you can, what you think will happen to you as [MS/control condition]."

## Dependent variables

We used several dependent measures typical to the TMT literature. The first two studies assessed one of the original measures in TMT research: the amount of bail participants would post for a (hypothetical) arrested prostitute [5]. Per the tenets of TMT, awareness of death should promote anti-prostitution attitudes given that prostitution allegedly challenges the American cultural worldview. All four studies shared a task in which participants evaluated the author of an anti-American essay (Study 5 also included a pro-American essay), the text of which was a direct copy of those used in seminal TMT work [e.g., 5]. This essay, ostensibly written by an immigrant, was critical of the United States. Participants evaluated the author of the essay via four questions (e.g., "How much do you like this author?" Each question pertaining to the essay was rated on a seven-point Likert-type scale, where lower scores reflect greater worldview defense. The four items were averaged into a single composite (all $\alpha$s > .91). Modification to this measure and description of additional measures are described where appropriate within each study. All pertinent materials for this study are available at http://tmt.missouri. edu/materials.html, or our online repository (link below), unless otherwise noted.

## Purpose of the studies

Below, we describe the general purpose, aims, and hypotheses of the studies that are included in this manuscript. Given that each was intended to be a replication and extension of basic MS work, and that basic replication did not emerge in each study, we do not provide detailed background and justification of each hypothesis (as this would be beyond the scope of the present paper), though more detail is available upon request.

**Studies 1 and 2.** The purpose of the first two studies was to examine the contribution of the need to belong to worldview defense. The hypothesis of study 1 was that two additional types of death—"social death" (loss of all relationships) and "the end of the world," whereby all

humanity ceases to exist—can comparably engender worldview defense as MS. The goal of Study 2 was to examine the potential effects of belongingness on worldview defense. The original hypotheses for Study 2 were that MS would beget worldview defense, anxiety, and curbed belongingness—the study additionally examined whether anxiety and belongingness each uniquely mediated the effect of MS on worldview defense.

**Studies 3 and 4.** The purpose of these studies was to further examine belongingness as a potential mediator on the effect of MS on worldview defense. Specifically, this study tested the hypotheses that MS would beget decreased belongingness, which would increase worldview defense. Study 3 also considered fear as an alternative mediator variable (i.e., MS induces fear, which promotes worldview defense), as well as considered a participant's baseline political beliefs (i.e., endorsement of American values), given that worldview defense in this study was operationalized as reactions to an anti-American essay. Study 4 included a long-term romantic relationship break-up as an alternative anxiety-related prime; it was hypothesized that relationship termination would result in comparable worldview defense to MS relative to a control condition. Belongingness and fear were both tested as potential mediator variables on the effect of MS and worldview defense (i.e., MS induces fear and thwarts belonginess, which kindle worldview defense).

**Study 5.** Study 5 had two goals. First, to examine whether uncertainty or social rejection could have an impact on endorsement of cultural world views in a manner like MS. Second, to examine experiential avoidance as a potential moderator of MS. It was hypothesized that an uncertainty prime would engender worldview defense comparably to a typical MS prime (in this case, worse evaluations of the anti-American essay *and* greater endorsement of a pro-American essay). We further hypothesized that measures of experiential avoidance (unwillingness to experience undesirable thoughts/feelings) would moderate the defense (i.e., increase endorsement of the anti-American author and rejection for the pro-American author).

## Transparency and openness

All raw data, survey materials, STATA 13 (for studies 1–4; StataCorp, College Station Texas), RStudio (mini meta-analysis; RStudio Team, Boston, MA) code, and SPSS (IBM inc., Armonk, NY) syntax (Study 5) used to report the analyses are available at the study's OSF page at https://osf.io/qda5b/ (DOI: 10.17605/OSF.IO/QDA5B). As mentioned, we did not pre-register any of the studies reported here because these studies were not intended to be direct replication studies. Only study five included a formal *a priori* power analyses in determining sample sizes. For studies 1–4, sample sizes were based on previous work and reflect the maximum number of respondents possible per the available budget. Each sample was sufficiently powered for the analyses conducted. Analysis and adequacy of sample sizes is presented in the discussion section.

## Data analysis overview

First, to address the main concerns of this paper, control groups and MS conditions were compared using Welch's *t*-tests to account for any inequality of variance and imbalance in group size; and all effect sizes *g* were adjusted for variance inequality. After, condition and age were into regression models predicting anti-American essay evaluations, followed by other pertinent covariates in subsequent models. All regression analyses were computed using robust sandwich-estimated standard errors to account for inequality of variance. In addition, since study 5 administered both a pro- and anti-American essays, we also present the originally planned 4(*Condition*: death, uncertainty, dental pain, TV) x 2(*Essay type*: pro- vs. anti-national) univariate ANOVA. The model was then repeated with all planned covariates entered as an ANCOVA.

Finally, in all exploratory moderation analyses below, the moderator variables were mean centered to facilitate interpretation of interactions. All additional analyses are described within each study for ease of presentation.

## Study 1

### Participants

A total of 233 participants initiated the task who were recruited from Amazon.com's MTurk (hereafter referred to as "MTurk"), a platform that facilitates crowd-sourced data and completion of brief tasks for a fee. The merits and limitations of this (and similar) platform are considered in the discussion. One respondent was removed from analyses for being under the age of 18, seven for indicating their answers should not be kept for analysis, and 22 for not providing responses to the two dependent measures. The final sample consisted of 203 participants (89 men). The age range of the sample was 18–76 years ($M = 38.23$, $SD = 13.70$). Participants were compensated $0.35.

### Procedure

Participants began by completing baseline psychosocial questionnaires. They then engaged in the MS prime, set bail for a hypothetical arrested prostitute, completed a questionnaire related to attitudes towards prostitution (described below) to examine if effects of MS appear only in those who oppose prostitution, the growing stone task (a neutral task that serves as a delay to allow the effects of MS to manifest), and then the evaluation of the anti-American essay.

**Writing primes and randomization.**   Based on random assignment, participants wrote about the end of the world and all human life ("all human life suddenly comes to an end;" $n = 47$), based on previous work [60], about losing all social bonds ("losing all of your friends and family;" $n = 60$), which was novel to this study to address the main hypotheses, a classic MS prime ($n = 55$), or about dental pain ($n = 41$). Because no differences emerged between the three non-MS conditions ($F$s < 1.86), they were combined into a single comparison condition ($n = 162$).

**Additional pre- and post-manipulation measures.**   Prior to the MS manipulation, participants completed the Rosenberg Self-Esteem Scale [RSES; 61], Single-Item Self-Esteem scale [SISE; 62], the Fear of Missing Out (FoMO) scale [63], the Ten-Item Personality Inventory [TIPI; 64], the 12-item General Belongingness Scale [GBS; 65], an eight-item measure rating endorsement of American values developed by the first author (e.g., *I endorse the values of the United States*") rated on a scale of 1 (*not true at all*)– 7 (*very true*) rated on an average ($\alpha$ = .76), and a six-item measure of mood that included three items averaged into an index of positive mood ("good," "happy," "cheerful;" $\alpha$ = .95) and three items averaged into an index of negative mood ("sad," "gloomy," and "depressed;" $\alpha$ = .94) rated on a scale of (*not at all*) to 7 (*extremely*).

Following the writing prime, participants completed a filler "Growing Stone" delay task [27] in which they read a brief passage, assessed whether the author was male or a female, and rated how descriptive they found the passage to be. Next, participants completed both the dependent measures and other inventories relevant to the purpose of each study and then responded to two questions to assess honesty: "It is very important that the answers obtained in this study entail honest responses. Please rate the degree to which you believe you were honest in the answers you had provided." The question was measured with the scale 1 (*Not honest at all*) to 7 (*Very honest*). Second, participants reported whether they believed their answers should be kept for analyses with a "yes/no" response. Participants who responded "no" to this question were omitted from the analyses. Finally, participants were debriefed on the purpose of the study and

provided relevant citations. Unless otherwise noted, all self-reported measures were assessed using the scale 1 (*Not true at all*) to 7 (*Very true*).

**Additional dependent measures.**   Prior to the essay evaluation, participants were asked to set bail for an arrested prostitute per previous work [5]. More specifically, participants read a passage about how bond is legally determined and then set bail between $0 and $1000 for a hypothetical person arrested for prostitution. Higher bail is operationally defined as greater worldview defense. Participants completed the bail task prior to essay evaluation. There were two central hypotheses for these studies: compared to the control essay, the MS condition would result in (a) higher bail for prostitutes and (b) more negative evaluations of the author of the anti-American essay.

Following the bail task, participants completed a measure of attitudes towards prostitution using three items from the Attitudes Toward Prostitution Scale (e.g., "Prostitution should be decriminalized") used in previous work [66]. The $\alpha$ was .92 ($M$ = 3.63, $SD$ = 2.05). The MS ($M$ = 3.61, $SD$ = 2.03) and the control ($M$ = 3.41, $SD$ = 1.98) conditions did not differ in this variable, $t$ (89.37) = 0.48, $g$ = 0.10, 95% $\underline{CI_g}$ = -0.30, 0.50, $p$ = .634. Participants rated the anti-American essay via four questions (e.g., "I agree with this person's opinion of America;" $\alpha$ = .95). Lower scores on this measure (i.e., lower approval of prostitution) represent higher worldview defense [5].

## Results

**Bail for prostitutes.**   The difference between the MS condition ($M$ = $383.25, $SD$ = $292.04) comparison conditions ($M$ = $328.25, $SD$ = $289.19) in the bail set for an arrested prostitute was not significant, $t$ (97.29) = 1.20, $p$ = .23, $g$ = 0.19, 95% $CI_g$ = (-0.12, 0.50). No support for the expected effect of MS emerged. A regression model with condition, age, and condition X age interaction was not significant, $F < 1.0$, $p > .5$, $R$ = 0.10, $R^2$ = 0.01.

**Evaluation of an anti-American essay.**   The MS conditions' ratings of the anti-American ($M$ = 3.55, $SD$ = 1.92) and the comparison conditions ($M$ = 3.56, $SD$ = 1.79) were nearly identical, $t$ (92.47) = 0.04, $p$ = .97, $g$ = 0.01, 95% $CI_g$ = (-0.30, 0.31). Again, these results do not support the MS condition impacted worldview. A regression model with age, condition, and the age X condition interaction was not significant, $F$ = 1.90, $p$ = .13, $R$ = 0.16, $R^2$ = 0.03. Despite a null main effect, in this model, age was negatively associated with essay evaluations, $b$ = -0.039, $SE$ = 0.02, 95% $CI_b$ = (-0.067, -0.01), $\beta$ = -.30, $p$ = .018. MS also resulted in more *positive* evaluations of the essay author than did the end of the world prime, $b$ = -2.14, $SE$ = 1.06, 95% $CI_b$ = (-4.23, -0.048), $\beta$ = -.27, $p$ = .045.

## Exploratory moderation analyses

We proposed four exploratory hypotheses pertaining to the moderating impact of attitudes towards prostitution, self-esteem, endorsement of American values, and religiosity.

**Attitudes towards prostitution.**   Baseline attitudes toward prostitution may account for the results seen in that task—specifically, we explored if the effect of MS on bail for persons of negative (vs. positive) attitudes towards prostitution was greater as prostitution may threaten their worldview. A regression analysis did not yield an interaction between the experimental condition and attitudes towards prostitution, $b$ = -19.28, $SE_b$ = 29.99, 95% $CI_b$ = (-78.84, 40.27), $\beta$ = -0.08, $p$ = .522. Unsurprisingly, more positive attitudes towards prostitution led to lesser bail, $b$ = -52.83, $SE_b$ = 21.10, 95% $CI_b$ = (-94.73, -10.94), $\beta$ = -0.35, $p$ = .014.

**Self-esteem.**   Worldview defense may occur for people of lower (vs. higher) self-esteem [67]. In the analyses of bail amount, no interaction between the experimental condition and self-esteem emerged, $b$ = -40.34, $SE_b$ = 35.97, 95% $CI_b$ = (-111.80, 31.13), $\beta$ = -0.16, $p$ = .265. Self-

esteem was also unassociated with bail amount set, $b = 11.25$, $SE_b = 23.41$, 95% $CI_b = (-35.26,$
57.76), $\beta = 0.07$, $p = .632$.

The analyses of the evaluation of anti-American essay yielded a statistically significant interaction, $b = 0.51$, $SE_b = 0.23$, 95% $CI_b = (0.05, 0.97)$, $\beta = 0.33$, $p = .031$. When decomposed, the interaction effect suggested that self-esteem was associated with more negative evaluations of the essay in the MS, $b = -0.35$, $SE = 0.14$, 95% $CI_b = (-0.62, -0.07)$, $p = .014$, versus in the comparison condition, $b = 0.16$, $SE = 0.18$, 95% $CI_b = (-0.21, -0.53)$, $p = .391$. Thus, this test reflects TMT's hypothesis that higher self-esteem decreased worldview defense following MS.

**Endorsement of American values.** The effect of MS on evaluation of the anti-American essay may be present in those who espouse (vs. do not espouse) American values, such as political conservatives. Since American values are specifically challenged in the essay, the interaction between the experimental condition and endorsement of American values was performed only for the evaluations of the essay. The evaluation was unaffected by the interaction between the experimental condition and endorsement of American values, $b = 0.44$, $SE = 0.38$, 95% $CI_b = (-0.31, 1.20)$, $\beta = 0.18$, $p = .247$. Endorsement of American values negatively correlated with evaluations of the essay, $b = -0.74$, $SE = 0.25$, 95% $CI_b = (-1.23, -0.25)$, $\beta = 0.33$, $p = .003$, positively to bail ($\beta = 0.36$, $p < .001$), although it did not interact with the experimental condition ($\beta = 0.02$, $p = .806$)

**Religiosity.** It is possible that religiosity buffers the effects of MS [68]. However, no interaction between the experimental condition and religiosity emerged for bail set for prostitutes, $b = 5.35$, $SE = 28.93$, 95% $CI_b = (-52.11, 62.81)$, $\beta = 0.03$, $p = .854$. The examination of the anti-American essay, however, yielded different results, $b = 0.45$, $SE = 0.17$, 95% $CI_b = (-0.11, 0.78)$, $\beta = 0.38$, $p = .01$. Higher religiosity led to more negative evaluations of the anti-American essay in the MS, $b = -0.27$, $SE = 0.11$, 95% $CI_b = (-0.49, -0.06)$, $p = .014$, but not in the control condition, $b = 0.18$, $SE = 0.13$, 95% $CI_b = (-0.08, 0.43)$, $p = .176$. These findings support the TMT-derived hypothesis that religiosity may bolster one's worldviews to buffer negative effects of MS, whereas the MS itself did not correspond to significant differences.

## Study 2

### Participants

A total of 124 participants were recruited from MTurk. We removed three participants from analyses for reporting that their answers should not be kept for analysis, and 18 for not completing either, or both, dependent measures. The total number of participants analyzed for this study was 103 (45 men, two who did not report their gender), with a mean age of 39.23 years ($SD = 14.26$, range = 18, 73; two did not report their age). The sample was 98% US citizen. Participants were compensated \$0.35 and were comparably split in the MS condition ($n = 52$) and the control condition ($n = 51$).

### Procedure

The procedures for Study 1 and 2 were nearly identical. Study 2 differed primarily in two ways: the MS induction and control conditions were streamlined into classic "death" vs. "dental pain" primes, and the included questionnaires differed (described further below).

### Additional pre- and post-manipulation measures

As in Study 1, participants began by completing the RSES [69] and the SISE [62], combined and averaged into an index of self-esteem ($\alpha = .94$); the FoMO scale [63]. The interdependence subscale of the Self-Construal Scale [70] and the TIPI [64] served as a filler measure.

Following the writing prime and the Growing Stone task, participants completed the General Belongingness Scale [GBS; 65] and a 16-item measure of mood used in prior work exploring affective reactions to MS [71]—these diverged from Study 1. Participants were then presented with the bail task followed by a measure of attitudes towards prostitution (identical to that of Study 1) for ancillary analyses ($\alpha = .91$; $M = 3.89$, $SD = 2.02$). They then completed the essay evaluation. Finally, participants completed a demographic inventory (gender, age, whether they were born in the U.S., whether they were a U.S. citizen, their education level, whether English is their native language) and reported their degree of political liberalism, conservatism, and independence (for ancillary analyses), the same eight-item measure of American values used in Study 1 ($\alpha = .80$), a measure of fear which was an average of four items reflecting fear-related emotions adapted from previous work [71] ("anxious," "fearful," "nervous," "afraid"; $\alpha = .90$), a three-item measure of religiosity created for the purposes of this study (e.g., "I am a religious person") averaged into a religiosity composite for the purposes of additional analyses ($\alpha = .99$), the time and day of the week that they completed the survey, and the same two questions assessing honesty described above.

## Results

**Bail for prostitutes.** The MS ($M = \$243.88$, $SD = \$36.18$) and the control conditions ($M = \$320.86$, $SD = \$37.16$) did not significantly differ in the bail set for the hypothetical arrested prostitute, $t$ (102.86) = -1.48, $p = .14$, $g = -0.29$ 95% $CI_g = (-0.68, 0.10)$. The results of this test did not change when bail was log-transformed.

**Evaluation of the anti-American essay.** The MS ($M = 2.90$, $SD = 1.61$) and control ($M = 3.06$, $SD = 1.71$) conditions did not differ in the evaluation of the anti-American essay, $t$ (101.64) = -0.49, $p = .62$, $g = -0.10$, 95% $CI_g = (-0.48, 0.29)$. The regression model with age and age X condition interaction was not significant, $F < 1.0$, $p = .80$, $R = 0.10$, $R^2 = 0.01$.

**Fear.** The MS ($M = 1.66$, $SD = 1.16$) and the control ($M = 1.99$, $SD = 1.37$) condition did not differ in fear, $t$ (99.55) = -1.31, $p = .19$, $g = -0.26$, 95% $CI_g = (-0.64, 0.13)$. The regression model with age and age X condition interaction was not significant, $F = 2.23$, $p = .090$, $R = 0.25$, $R^2 = 0.06$. Given the goals of the present paper, we also examined the coefficients despite the null model. The coefficients suggested a significant effect of condition such that the MS primes resulted in greater fear than the dental pain condition, $\beta = 0.25$, $p = .018$. The age X condition interaction term was also significant, $b = -0.04$, $SE_b = .02$, 95% $CI_b = (-0.07, -0.00)$, $\beta = -.42$, $p = .038$, such that as age increased, fear in the death condition increased and fear in the dental pain condition decreased, counter to hypotheses previous literature on fear from death thoughts [e.g., 72]. Dental fear does, generally, decrease with age [73].

## Exploratory moderation analyses

As in Study 1, we conducted exploratory moderation analyses to determine whether individual differences in psychosocial variables influenced the results. We examined attitudes towards prostitution, self-esteem, endorsement of American values, political orientation, and religiosity as moderating variables. We mean-centered any continuous moderator.

**Attitudes towards prostitution.** Unsurprisingly, attitudes towards prostitution were negatively associated with bail (i.e., more positive attitudes led to lesser bail, $b = -40.87$, $SE_b = 20.06$, 95% $CI_b = (-80.68, -1.06)$, $\beta = -0.31$, $p = .044$. However, attitudes towards prostitution did not interact with the experimental condition to predict bail in a regression analysis, $b = 22.37$, $SE_b = 27.10$, 95% $CI_b = (-31.40, 76.14)$, $\beta = 0.12$, $p = .411$. To reduce the likelihood of Type I error and because we do not believe the relation between the variables would be

meaningful, we did not analyze the relation of attitudes toward prostitution with the essay evaluation and fear.

**Self-esteem.** A regression analysis with bail as the dependent variable revealed no significant main effect of self-esteem, $b = 30.22$, $SE_b = 27.40$, 95% $CI_b = (-24.14, 84.59)$, $\beta = 0.16$, $p = .273$, nor an interaction between self-esteem and experimental condition, $b = 14.56$, $SE_b = 35.91$, 95% $CI_b = (-56.69, 85.82)$, $\beta = 0.05$, $p = .686$. Log transforming bail did not change these results. We also ran a regression with a three-way interaction between condition, attitudes towards prostitution, and self-esteem. TMT would suggest that negative attitudes towards prostitution leads to higher bail only among persons with low self-esteem. Although this model produced a statistically significant three-way interaction, the results were mixed and generally counter to predictions. For example, persons of high self-esteem and positive attitudes towards prostitution reported higher bail ($M = \$709.50$) than persons of lowest self-esteem and lowest acceptability of prostitution ($M = \$546.50$). The interaction between experimental condition and attitudes towards prostitution remained null. These results did not change with log-transformed bail.

A regression analysis with the ratings of the anti-American essay as the dependent variable revealed a significant main effect of self-esteem such that persons of higher (vs. lower) self-esteem were less favorable of the author of the essay, $b = -0.33$, $SE_b = 0.14$, $CI_b = -0.61, -0.04$, $\beta = -0.28$, $p = .025$. However, there was no significant interaction effect between the experimental condition and self-esteem, $b = -0.12$, $SE_b = 0.22$, $CI_b = (-0.57, 0.32)$, $\beta = -0.07$, $p = .589$.

When fear was the dependent variable, a negative effect of self-esteem emerged, $b = -0.31$, $SE_b = 0.13$, 95% $CI_b = (-0.57, -0.05)$, $\beta = -0.35$, $p = .018$, such that participants who reported higher (vs. lower) self-esteem reported less fear overall. Self-esteem did not interact with the experimental condition in predicting fear, $b = -0.18$, $SE_b = 0.18$, 95% $CI_b = (-0.54, 0.18)$, $\beta = -0.13$, $p = .320$.

**Political orientation.** A positive trend emerged between conservatism and bail, $b = 33.86$, $SE_b = 17.20$, 95% $CI_b = (-0.28, 68.00)$, $\beta = 0.26$, $p = .052$. Liberalism was negatively associated with bail, $b = -37.73$, $SE_b = 14.29$, 95% $CI_b = (-66.09, -9.37)$, $\beta = -0.31$, $p = .010$. In other words, as liberalism increased, the set bail decreased, and the inverse was true for conservativism. Independence was unrelated to bail, $b = -4.49$, $SE_b = 17.83$, 95% $CI_b = (-39.88, 30.90)$, $\beta = -0.03$, $p = .802$. The experimental condition did not interact with conservatism, $b = 1.02$, $SE_b = 26.01$, 95% $CI_b = (-50.60, 52.64)$, $\beta = 0.01$, $p = .969$, liberalism, $b = 23.43$, $SE_b = 23.62$, 95% $CI_b = (-23.46, 70.32)$, $\beta = 0.13$, $p = .324$, or independence, $b = -2.30$, $SE_b = 28.97$, 95% $CI_b = (-59.81, 55.21)$, $\beta = -0.01$, $p = .937$, in predicting bail. These results did not change with log transformed bail. Including self-esteem neither changed the results nor yielded a three-way interaction. Adding attitudes towards prostitution did not produce a three-way interaction.

The main effects of conservatism, $b = -0.37$, $SE_b = 0.08$, 95% $CI_b = (-0.53, -0.22)$, $\beta = -0.46$, $p < .001$, liberalism, $b = 0.31$, $SE_b = 0.10$, 95% $CI_b = (0.10, 0.52)$, $\beta = 0.41$, $p = .004$, and independence, $b = -0.03$, $SE_b = 0.13$, 95% $CI_b = (-0.30, 0.23)$, $\beta = -0.04$, $p = .804$, mirrored their correlations with the essay evaluation. However, there was no interaction with the experimental condition in predicting evaluations of the essay for conservatism, $b = 0.28$, $SE_b = 0.15$, 95% $CI_b = (-0.01, 0.58)$, $\beta = 0.26$, $p = .062$, liberalism, $b = -0.27$, $SE_b = 0.16$, 95% $CI_b = (-0.58, 0.04)$, $\beta = -0.25$, $p = .089$, and independence, $b = 0.15$, $SE_b = 0.19$, 95% $CI_b = (-0.22, 0.53)$, $\beta = 0.13$, $p = .409$. Adding self-esteem yielded a three-way interaction between experimental condition and political orientation only for conservatism. When the interaction was deconstructed, it revealed that self-esteem predicted essay evaluations in the dental pain condition, $\beta = -0.44$, $p = .005$, and not the MS condition $\beta = -0.19$, $p = .12$. It also indicated that conservatism predicted evaluations of the essay in the experimental condition $\beta = -0.46$, $p < .001$, but not the

control condition $\beta = 0.04$, $p = .767$. Given that these results do not follow a pattern based on theory, we interpret them as spurious.

In the examination of fear, conservatism, $b = 0.08$, $SE_b = 0.11$, 95% $CI_b = (-0.15, 0.31)$, $\beta = 0.09$, $p = .487$, liberalism, $b = 0.07$, $SE_b = 0.12$, 95% $CI_b = (-0.16, 0.30)$, $\beta = 0.08$, $p = .558$, and independence, $b = 0.18$, $SE_b = 0.13$, 95% $CI_b = (-0.07, 0.43)$, $\beta = 0.20$, $p = .149$, yielded no significant interaction effects with the experimental condition. Independence was unassociated with fear, $b = -0.13$, $SE_b = 0.08$, 95% $CI_b = (-0.07, 0.43)$, $\beta = -0.21$, $p = .113$, whereas liberalism was positively associated with it, $b = 0.19$, $SE_b = 0.08$, 95% $CI_b = (0.02, 0.36)$, $\beta = 0.32$, $p = .029$, and conservatism was negatively associated with it, $b = -0.14$, $SE_b = 0.07$, 95% $CI_b = (-0.27, -0.01)$, $\beta = -0.22$, $p = .040$. Including an interaction between self-esteem, experimental condition, and political orientation did not result in significant two- or three-way interactions.

**Endorsement of American values.** This analysis was conducted as it is critical of American values. No interaction effects emerged between the experimental condition and endorsement of American values in predicting evaluations of the essay, $b = 0.36$, $SE_b = 0.28$, 95% $CI_b = (-0.20, 0.92)$, $\beta = 0.17$, $p = .20$. The main effect of experimental condition continued to be null, $b = 0.24$, $SE_b = 0.30$, 95% $CI_b = (-0.36, 0.84)$, $\beta = 0.07$, $p = .43$. Unsurprisingly, endorsement of American values was negatively related to evaluations of the essay author, $b = -0.65$, $SE_b = 0.13$, 95% $CI_b = (-0.90, -0.40)$, $\beta = -0.49$, $p < .001$.

Self-esteem did not yield a three-way interaction with experimental condition and endorsement of American values when added into the model. The interaction between condition and endorsements of American values was significant, $b = 0.63$, $SE = 0.30$, 95% $CI_b = (0.04, 1.23)$, $\beta = 0.31$, $p = .04$. When decomposed, the pattern of differences was consistent with TMT's predictions: controlling for self-esteem and endorsement of American values negatively predicted evaluations of the essay in the experimental ($\beta = -0.51$, $p < .001$) but not the control condition ($\beta = 0.02$, $p = .90$).

**Religiosity.** Religiosity did not interact with the experimental condition in predicting bail, $b = 7.37$, $SE_b = 23.11$, 95% $CI_b = (-38.48, 53.23)$, $\beta = 0.05$, $p = .75$, essay evaluation, $b = 0.19$, $SE_b = 0.15$, 95% $CI_b = (-0.11, 0.48)$, $\beta = 0.20$, $p = .21$, or fear, $b = 0.08$, $SE_b = 0.09$, 95% $CI_b = (-0.10, 0.26)$, $\beta = 0.11$, $p = .36$. The effect of religiosity was also null for both bail, $b = 18.52$, $SE_b = 17.17$, 95% $CI_b = (-15.56, 52.61)$, $\beta = 0.17$, $p = .28$, and for essay evaluation, $b = -0.14$, $SE_b = 0.10$, 95% $CI_b = (-0.35, 0.07)$, $\beta = -0.20$, $p = .19$. Religiosity was negatively associated with fear, $b = -0.13$, $SE_b = 0.06$, 95% $CI_b = (-0.24, -0.01)$, $\beta = -0.24$, $p = .03$.

Adding self-esteem as a predictor yielded a three-way interaction for evaluations of the anti-American essay. When decomposed, this effect suggests that an interaction between religiosity and self-esteem emerged only in the experimental condition, $b = 0.13$, $SEb = 0.06$, 95% $CI_b = (0.02, 0.25)$, $\beta = 0.29$, $p = .002$. When decomposed further, this effect follows a trend parallel to TMT's predictions: in the experimental condition, religiosity predicts unfavorable attitudes towards the essay's author when self-esteem is below the mean of the sample ($b = -0.42$, $SE_b = 0.12$, 95% $CI_b = (-0.66, -0.18)$, $\beta = -0.48$, $p = .002$) versus above ($\beta = 0.16$, $p = .59$).

## Study 3

### Participants

A total of 285 participants began this study: 279 respondents from MTurk and nine from the *samplesize* "subreddit," or online forum within reddit.com (www.reddit.com/r/samplesize). We removed six participants from analyses for reporting that their answers should not be kept for analysis and an additional 49 for failing to complete the dependent measures. The final sample consisted of 230 participants (83 men; one person did not report gender), of which 110 participants were assigned to the MS condition, and 120 participants were assigned to the

control condition (dental pain). The mean age of the sample was 35.55 years ($SD$ = 11.62, range = 18–74).

## Pre- and post-manipulation measures

Before completing the essays, participants provided demographic variables: gender, age, US citizenship status, and, if not a US citizen, how long they have lived in the United States, the website where they found the link to the survey, political party identification, degree of liberalism/conservatism in social and fiscal issues (these last three items were averaged into an overall composite of liberalism/conservatism to be used for ancillary analyses whereby higher values reflect more conservative views; α = .88), a five-item set of questions designed by the first author to assess general endorsement of American values, similar to those used in Studies 1 and 2: "I endorse the values of the United States," "The United States provides more opportunities for people than does any other country," "I hold American values close to me," "America is the greatest nation in the world," and "American values are inferior compared to those of other countries" (the first four items were averaged into an overall composite of American value endorsement for ancillary analyses because the final item was unrelated to the rest; α = .92), religiosity ("I see myself as someone who is very religious;" used for ancillary analyses below), TIPI , and RSES for ancillary analyses (α = .93). After essay evaluation, participants completed the GBS [65], the SISE [62], a measure of fear described above (composite of *anxious*, *fearful*, *nervous*, and *afraid* ; α = .90), and the two honesty items described above.

## Results

**Tests of study's original hypothesis.** Belongingness did not differ between the MS and the control conditions, $t$ (226.96) = 0.42, $g$ = 0.06, 95% $CI_g$ = (-0.20, 0.31), $p$ = .68. Belongingness was unrelated to essay evaluation, $r$ = .03, $p$ = .64. Adding age did not improve model fit, $F$ < 0.9, $p$ = .46. Due to these null relations, belongingness did not qualify as a potential mediator variable behind MS's effect on worldview defense.

**Essay evaluation.** Participants in the control condition ($M$ = 3.60, $SD$ = 1.45) reported a less favorable view of the author than those in the MS condition ($M$ = 4.14, $SD$ = 1.62), which was statistically significant, $t$ (221.52) = 2.67, $g$ = 0.35, 95% $CI_g$ = (0.09, 0.61), $p$ = .009. Note that this result was in the opposite direction of hypotheses: we would expect the MS condition to rate the evaluation *less* favorably than the control condition. Adding age to the model resulted in a significant model fit, $F$ (3, 226) = 4.05, $p$ = .008, $R$ = 0.23, $R^2$ = 0.05. However, the main effects of condition, age, and the condition X age interaction term were not significant, $t$s < 1.1, $b$s < 0.09, $p$s > .28. Therefore, the model including age is uninterpretable.

**Fear.** The MS ($M$ = 2.30, $SD$ = 1.47) and the control conditions ($M$ = 2.12, $SD$ = 1.40) did not significantly differ in fear, $t$ (220.73) = 0.95, $g$ = 0.13, 95% $CI_g$ = (-0.13, 0.39), $p$ = .34. Including age resulted in a null model, $F$ = 1.75, $p$ = .158, $R$ = 0.15, $R^2$ = 0.02.

## Exploratory moderation analyses

Like Study 2, self-esteem, political orientation, religiosity, and endorsement of American values were considered as variables with potential to moderate the effects of MS.

**Self-esteem.** Self-esteem did not interact with essay evaluation, $b$ = -0.10, $SE_b$ = 0.17, 95% $CI_b$ = (-0.44, 0.23), $β$ = -0.06, $p$ = .55. Self-esteem was also statistically unrelated to evaluations of the essay, $b$ = -0.20, $SE_b$ = 0.13, 95% $CI_b$ = (-0.46, 0.05), $β$ = -0.17, $p$ = .12. Self-esteem did not interact with the experimental condition to predict fear, $b$ = 0.08, $SE_b$ = 0.15, 95% $CI_b$ = (-0.21, 0.37), $β$ = 0.05, $p$ = .57. Self-esteem was negatively associated with fear, $b$ = -0.57, $SE_b$ = 0.10, 95% $CI_b$ = (-0.77, -0.38), $β$ = -0.52, $p$ < .001.

**Political orientation.** The interaction between political orientation and the experimental condition was null for both evaluations of the essay's author, $b = -0.01$, $SE_b = 0.14$, 95% $CI_b = -(0.29, 0.07)$, $\beta = -0.01$, $p = .94$) and for fear, $b = 0.04$, $SE_b = 0.15$, 95% $CI_b = (-0.25, 0.32)$, $\beta = 0.03$, $p = .81$. Political orientation was negatively associated with essay evaluations, whereby more conservatism leads to more negative evaluations of the essay's author ($b = -0.28$, $SE_b = 0.11$, 95% $CI_b = (-0.49, -0.07)$, $\beta = -0.28$, $p = .01$), and was unrelated to fear, $b = -0.10$, $SE_b = 0.12$, 95% $CI_b = (-0.33, 0.12)$, $\beta = -0.11$, $p = .37$.

When we added self-esteem to the model, a three-way *condition* x *political orientation* x *self-esteem* interaction emerged, $b = -0.40$, $SE_b = 0.11$, 95% $CI_b = (-0.62, -0.17)$, $\beta = -0.33$, $p = .001$. When decomposed, an effect emerged only for persons in the bottom 25th percentile of self-esteem scores ($N = 51$), $b = 0.55$, $SE_b = 0.27$, 95% $CI_b = (0.02, 1.09)$, $\beta = 0.34$, $p = .04$. Among those with relatively low self-esteem, political conservatism was associated with unfavorable evaluations of the essay's author in the mortality salience, $r = -.59$, $p < .001$, but not in the dental pain condition, $r = -.07$, $p = .76$. Further, self-esteem did not yield a three-way interaction when added into the model and no other interactions emerged.

**Endorsement of American values.** Like Study 2, endorsement of American values was entered as a moderator for evaluations of the essay, though no interaction effect emerged, $b = -0.10$, $SE_b = 0.12$, 95% $CI_b = (-0.33, 0.14)$, $\beta = -0.08$, $p = .42$. Higher endorsement of American values did lead to more negative evaluations of the essay's author, $b = -0.36$, $SE_b = 0.09$, 95% $CI_b = (-0.55, -0.18)$, $\beta = -0.39$, $p < .001$. The inclusion of self-esteem yielded a three-way interaction between the experimental condition, endorsement of American values, and self-esteem, $b = -0.21$, $SE_b = 0.28$, 95% $CI_b = (0.38, -0.05)$, $\beta = -0.23$, $p = .01$. When decomposed, we found that an interaction between the experimental condition and self-esteem emerged only for those of relatively high endorsement of American values ($n = 44$), $b = -0.87$, $SE_b = 0.40$, 95% $CI_b = (-1.68, -0.07)$, $\beta = -0.48$, $p = .03$. We did not probe this interaction further due to the small size of this sub-sample.

*Religiosity*. No interaction effect between the experimental condition and religiosity emerged when examining evaluations of the essay's author, $b = -0.08$, $SE_b = 0.09$, 95% $CI_b = -(0.26, 0.10)$, $\beta = -0.08$, $p = .37$. Religiosity was also unrelated to essay evaluation, $b = 0.05$, $SE_b = 0.07$, 95% $CI_b = (-0.18, 0.09)$, $\beta = -0.07$, $p = .50$. No interaction effect emerged between the experimental condition and religiosity when predicting fear, $b = -0.05$, $SE_b = 0.09$, 95% $CI_b = -(0.24, -0.13)$, $\beta = -0.06$, $p < .585$. Religiosity was unrelated to fear, $b = 0.01$, $SE_b = 0.07$, 95% $CI_b = (-0.12, 0.15)$, $\beta = 0.02$, $p = .85$.

## Study 4

### Participants

Participants were 227 adults recruited from MTurk. We removed 11 for not responding that their answers be kept for analysis and 38 for not completing the writing prime. Thus, the final sample consisted of 178 participants (81 men; one person did not report gender). The mean age of the sample was 36.80 years ($SD = 11.82$, range = 18–79).

### Writing primes and randomization

This study included three conditions: (a) the MS prime ($n = 66$); (b) break-up with a long-term romantic partner prime ($n = 60$); and (c) a control prime in which participants wrote about browsing the internet ($n = 52$). We combined the internet browsing and the relationship loss primes were into a single comparison condition ($n = 112$) because they did not significantly differ on any outcome measure ($ts < 1.53$).

### Pre- and post-manipulation measures

Prior to the presentation of the prime, participants reported their age, gender and completed questionnaires. Questionnaires included: (a) US citizenship status or length of residency in the US if they were not citizens, (b) a 5-item measure of American values (identical to Study 3; $\alpha$ = .85), (c) general political orientation (1 = *Very liberal;* 2 = *Quite liberal;* 3 = *Somewhat liberal;* 4 = *Moderate*; 5 = *Somewhat conservative;* 6 = *Quite conservative;* and 7 = *Very conservative*), (d) measures of political orientation for social (e.g., marriage equality) and economic issues (e.g., government spending), measured on the same scale of 1–7 as general political orientation (these three items were averaged into a single composite political orientation score; $\alpha$ = .89), and the RSES [61] ($\alpha$ = .92).

After the "Growing Stone" essay evaluation, participants completed a 44-item measure of the Big Five as a filler measure [74], completed the essay evaluation task identical to the studies above, the GBS [65] ($\alpha$ = .86), the SISE [62], and a 16-item measure of emotions that was used to compute a composite of fear: anxious, fearful, afraid, and nervous, as above ($\alpha$ = .92).

### Results

**Evaluation of the anti-American essay.**   The MS condition ($M$ = 3.97, $SD$ = 1.54) did not significantly differ from the comparison ($M$ = 3.65, $SD$ = 1.62) in the evaluation of the anti-American essay, $t$ (143.59) = 1.31, $g$ = 0.20, 95% $CI_g$ = (-0.10, 0.50), $p$ = .19. Including age, condition, and the age X condition interaction term in a regression resulted a significant model, $F$ (3, 174) = 3.1, $p$ = .006; $R$ = 0.26, $R^2$ = 0.07. Examination of the coefficients indicated that, as age increased, negative evaluations of the author decreased, $b$ = -0.03, $SE_b$ (0.01), 95% $CI_b$ (-0.05, -0.00), $t$ = -2.10, $p$ = .037, $\beta$ = -.19. The main effect of condition, and the interaction of age X condition, was not significant in this model ($ts$ < 1.6, $ps$ >.12).

**Fear.**   No difference in fear emerged between the MS ($M$ = 2.04, $SD$ = 1.42) and the comparison condition ($M$ = 2.06, $SD$ = 1.33), $t$ (131.09) = 0.11, $g$ = 0.02 95% $CI_g$ = (-0.29, 0.31), $p$ = .92. Like the essay evaluations, including age, condition, and the age X condition interaction term in a regression resulted a significant model, $F$ (5, 172) = 2.70, $p$ = .022; $R$ = 0.270, $R^2$ = 0.073. As with the essay evaluations, as age increased, negative evaluations of the author decreased, $b$ = -0.033, $SE_b$ (0.014), 95% $CI_b$ (-0.06, -0.01), $t$ = -2.37, $p$ = .019. The main effect of condition, and the interaction of age X condition, was not significant in this model ($ts$ < 1.0, $ps$ >.47).

### Exploratory moderation analyses

Self-esteem (measured prior to the manipulation), political orientation, religiosity, and endorsement of American values were again considered as variables with potential to moderate the effects of MS.

**Self-esteem.**   A regression analysis revealed that self-esteem did not yield an interaction effect with the experimental condition in the analysis of essay evaluation, $b$ = -0.001, $SE_b$ = 0.20, 95% $CI_b$ = (-0.40, 0.40), $\beta$ = .001, $p$ = .99. Self-esteem was unrelated to evaluations of the essay, $b$ = -0.23, $SE_b$ = 0.16, 95% $CI_b$ = (-0.54, 0.08), $\beta$ = -0.19, $p$ = .14. Self-esteem was negatively associated with fear, $b$ = -0.51, $SE_b$ = 0.15, 95% $CI_b$ = (-0.81, -0.21), $\beta$ = -0.48, $p$ = .001 and did not interact with the experimental condition to predict fear, $b$ = -0.01, $SE_b$ = 0.17, 95% $CI_b$ = -(0.35, 0.33), $\beta$ = -0.01, $p$ = .96.

**Political orientation.**   The interaction between political orientation and the experimental condition was null for evaluations of the essay's author, $b$ = 0.11, $SE_b$ = 0.13, 95% $CI_b$ = (-0.14, 0.36), $\beta$ = 0.06, $p$ = .40). As conservatism increased, so did negative evaluations of the essay's

author, $b$ = -0.54, $SE_b$ = 0.10, 95% $CI_b$ = (-0.75, -0.34), $\beta$ = -0.51, $p < .001$. Self-esteem did not interact with any variables when included in the model.

Political orientation displayed a moderating effect with the experimental condition in the analysis of fear, $b$ = - 0.24, $SE_b$ = 0.12, 95% $CI_b$ = (-0.48, -0.003), $\beta$ = -0.21, $p$ = .05. Simple slopes analyses revealed that political orientation was negatively associated with fear for the control condition, $b$ = -0.26, $SE_b$ = 0.10, 95% $CI_b$ = (-0.46, -0.07), $p$ = .01, but not for the MS condition, $b$ = 0.05, $SE_b$ = 0.09, 95% $CI_b$ = (-0.12, 0.22), $p$ = .58. Self-esteem interacted with the experimental condition and political orientation when included in the model, $b$ = 0.19, $SE_b$ = 0.09, 95% $CI_b$ = (0.02, 0.36), $p$ = .03. When decomposed, self-esteem and political orientation yield an interaction effect in the mortality salience, $b$ = -0.19, $SE_b$ = 0.07, 95% $CI_b$ -0.32, -0.06, $\beta$ = -0.22, $p$ = .01, but not the control condition, $b$ = -0.003, $SE_b$ = 0.06, 95% $CI_b$ = (-0.11, 0.11), $\beta$ = -0.004, $p$ = .96. Specifically, political conservatism was associated with fear among only those self-esteem scores below the mean, $n$ = 23; $\beta$ = 0.33, $p$ = .05, but not for persons with self-esteem scores above the mean: $n$ = 43; $\beta$ = 0.03, $p$ = .67.

**Endorsement of American values.**   No interaction effect emerged between the prime condition and endorsement of American values, $b$ = -0.08, $SE_b$ = 0.14, 95% $CI_b$ = (-0.35, 0.19), $\beta$ = -0.04, $p$ = .57. Unsurprisingly, greater endorsement of American values led to more negative evaluations of the essay's author, $b$ = -0.54, $SE_b$ = 0.11, 95% $CI_b$ = (-0.77, -0.32), $\beta$ = -0.50, $p < .001$. Including self-esteem in the model did not change these results.

The analysis of fear revealed an interaction between the prime condition and endorsement of American values, $b$ = -0.32, $SE_b$ = 0.13, 95% $CI_b$ = (-0.57, -0.07), $\beta$ = -0.27, $p$ = .01. Simple slopes analyses revealed that the endorsement of American values was associated with fear in the control prime condition, $b$ = -0.26, $SE_b$ = 0.08, 95% $CI_b$ = (-0.42, -.10), $p$ = .002, but not in the MS prime condition, $b$ = 0.06, $SE_b$ = 0.09, 95% $CI_b$ = (-0.13, 0.25), $p$ = .53. When included in the model, self-esteem did not interact with any variable.

**Religiosity.**   No interaction effect between the experimental condition and religiosity emerged when examining evaluations of the essay's author, $b$ = 0.06, $SE_b$ = 0.13, 95% $CI_b$ = -(0.32, -0.20), $\beta$ = -0.05, $p$ = .65. Religiosity was also unrelated to essay evaluation, $b$ = -0.19, $SE_b$ = 0.10, 95% $CI_b$ = (-0.40, 0.01), $\beta$ = -0.25, $p$ = .06. Although including self-esteem in this test yielded a self-esteem X religiosity interaction, it was beyond the scope and relevance of this investigation to probe this interaction further.

The experimental condition did not interact with religiosity in predicting fear, $b$ = -0.003, $SE_b$ = 0.10, 95% $CI_b$ = (-0.19, 0.19), $\beta$ = 0.004, $p$ = .98. Religiosity was unrelated to fear, $b$ = -0.04, $SE_b$ = 0.08, 95% $CI_b$ = (-0.20, 0.11), $\beta$ = -0.06, $p$ = .61. The addition of self-esteem into the model did not yield a three-way interaction nor any other interaction effects.

## Study 5

### Participants

A total of 352 participants consented to participate and moved to questionnaires. The study was posted on several websites, including the last author's lab web page, Psychological Research on the Net, and a list of psychology experiments posted on the University of Zurich's webpage. Although the latter two were hosted in European countries, they post studies from across the world and are frequently accessed by individuals in the United States. Participants were able to access a secure link to the study from these websites. In addition, students were recruited from introductory psychology courses from a large urban university in the Mid-Atlantic, United States who completed the study in exchange for course credit.

We removed 107 individuals for incomplete data (missing at least one questionnaire or key demographic information, such as country of birth. Retaining non-completers, who were US

residents but missed at least one questionnaire other than the essay evaluation, did not change significance or magnitude of the effects described below. Of the remaining 245 completers, we included only US residents aged 18 years old or older ($n = 183$; 44 men). Where reported, non-completers and completers were matched on gender ($p = .98$, $V < .01$), age ($p = .092$, $d = .21$), and essay randomization ($p = .29$, $V = .07$). Omitted participants were overrepresented in the TV-salience condition, $\chi^2$ (3) = 13.79, $p = .003$, $V = .24$, but were otherwise comparably distributed across the other conditions. Of these, 63% completed the questionnaire in the lab and whereas the remainder completed the study online (see procedures). Included participants ranged in age from 18 to 70 years (M = 24.52 years, $SD = 10.98$). Internet participants ($n = 66$) were significantly older (Mdn = 27, IQR = 16.00, M = 31.65, SD = 13.91) than in-lab participants ($n = 117$; Median = 19.00, IQR = 2.00, M = 20.51, SD = 5.97), Mann-Whitney U = 6211.00, $p < .001$, $r = .41$. Not surprisingly given the methods of recruitment, in-lab respondents were overrepresented as current undergraduates, $\chi^2$ (1) = 32.51, $p < .001$, $V = .42$. The two groups did not significantly differ in any other variable described below, $ts < 1.7$, $ps > .10$. They also were comparably matched in terms of gender and ethnicity, $ps > .07$, $Vs < .18$. Based on self-report, about 80% of participants were White, 11% were African American, 2% were Hispanic, 2% were Asian, 3% were multiracial or biracial, and about 1% did not report their ethnicity. The majority (85%) of the sample were undergraduates. Given these comparable distributions across groups, and well-established empirical equivalence between in-person vs. computer-administered social science research [75], we combined Internet and in-person respondents.

## Writing primes and randomization

Participants were randomly assigned to one of four independent variables (mortality salience, uncertainty salience, social pain, and watching TV) and to read a letter ostensibly written by a recent immigrant to the United States (pro- vs. anti-American) for which they provided ratings of the essay. The distribution of condition into the pro-American (mortality salience $n = 26$, uncertainty $n = 22$, social pain $n = 22$, and TV $n = 23$) and anti-American (mortality salience $n = 21$, uncertainty $n = 24$, social pain $n = 21$, and TV $n = 24$) were statistically comparable, $\chi^2$ (3) = .614, $p = .893$, $V = .058$. The four prime groups did not significantly differ in age or questionnaire responses described below, $Fs < 2.40$, $ps > .072$, $\omega^2 < .039$, or distribution of gender and ethnicity, $\chi^2 < 3.50$, $ps > .32$, $V < .14$.

## Additional pre- and post-manipulation measures

Prior to the manipulation, participants completed: (a) the RSES [61] as described above ($\alpha = .88$); (b) the 21-item Feagin's Intrinsic-Extrinsic Religiosity Orientation Scale [FIERS; 76], analyzed as a sum score for a composite of religiosity ($\alpha = .85$), (c) 36-item Right-Wing Authoritarian Scale [RWAS; 77], which measures an individual's preference for authoritarian politics ($\alpha = .76$); (d) the ten-item Acceptance and Action Questionnaire [AAQ; 78], which assesses experiential avoidance, where larger scores indicate reduced acceptance and increased avoidance ($\alpha = .76$), (e) the 21-item Depression, Anxiety, and Stress Scale [DASS-21; 79], which measures present feelings of depression, anxiety, and stress (each on its owns eponymous subscale) on a scale ranging from 0–3 (all $\alpha > .85$), (f) the 10-item, second form of the revised Marlowe-Crown Social Desirability Scale [MC 2; 80], which measures proneness to "fake good" by selecting 10 unrealistically prosocial behaviors answered as true or false (KR-20 = .57).

To allow for a delay and distraction following the prime, participants completed the 20-item Positive and Negative Affect Scale [PANAS-X; 81]. Participants rate negative or positive items

pertaining to mood. Separate scales of positive and negative affect are provided ($\alpha > .85$). This measure provides adequate delay between primes and the dependent variable [19, 23].

## Dependent variables

In addition to the anti-American essay (identical to studies 1–4), participants were also randomly assigned a pro-American essay identical to the ones used in the original TMT work [82].

## Procedures

Students who participated in-person came to the research laboratory and were seated in a small room alone. Remote participants completed the task on their own devices. After completing consent, participants completed the remaining procedures: questionnaires, writing prime, PANAS, then essay evaluation. After completing procedures, students were reimbursed course credit. Internet respondents were not reimbursed.

## Results

To address the main goals of the present paper, we collapsed the three control conditions (TV, social pain, and uncertainty) and compared their ratings of the essays to the mortality conditions. For the anti-American essay, the mortality group ($n = 21$) evaluated the writer more negatively ($M = 4.88$, $SD = 2.10$) than the control groups ($n = 69$; $M = 5.18$, $SD = 1.90$), though this did not approach significance, $t(88) = 0.62$, $p = .537$, $g = .15$. For the pro-American essay, the mortality group ($n = 26$), *also* rated the essay lower ($M = 5.97$; $SD = 1.48$) than the control groups ($n = 67$; $M = 6.17$; $SD = 1.32$), which, again, did not approach significance $t(91) = 0.64$, $p = .525$, $g = .15$. Thus, the direction one would expect for MS primes was negligibly observed in one condition (anti-American) and was in the wrong direction for the other (pro-American). Next, we examined the TV control conditions to the mortality conditions as TV can be considered a "purer" control and that there was notable variance across control groups (described below), we conducted this further looking at *just* the mortality conditions vs. the TV conditions evaluating the essays ($n_{\text{anti-American}} = 24$; $n_{\text{pro-American}} = 23$; the $n$s, means, and SDs of the mortality groups are unchanged). For the anti-American essay, the MS condition rated the essay more negatively than the TV condition ($M = 5.53$, $SD = 1.84$), though this did not approach significance, $t(43) = 1.12$, $p = .268$, $g = .33$. As above, and counter to expectations, the MS condition rated the pro-American essay *less* favorably than the TV condition ($M = 6.92$, $SD = 0.71$), which did rise to statistical significance, $t(47) = 2.82$, $p = .007$, $g = .80$.

**Test of original hypotheses.**   Results of the originally intended 4 (*Prime*: mortality vs. uncertainty vs. social pain vs. TV) x 2 (*Essay Type*: pro- vs. anti-American) ANOVA reflected the *t*-tests above. There was a main effect of essay type, $F(1,175) = 17.13$, $p < .001$, $\eta_p^2 = .09$, whereby the pro-American essay was rated more positively ($M = 6.11$, $SE = 0.17$) than the anti-American essay ($M = 5.11$, $SE = 0.17$). There was also a main effect of condition, $F(3,175) = 2.94$, $p = .035$, $\eta_p^2 = .048$. Post-hoc tests with least significant differences showed that the TV condition rated their essay significantly more positively ($M = 6.22$, $SE = 0.24$) than the mortality ($M = 5.42$, $SE = 2.4$), uncertainty ($M = 5.38$, $SE = 0.24$), and social pain ($M = 5.40$, $SE = 0.25$) conditions (all $ps < .05$). The use of Tukey or Bonferroni correction resulted in entirely null post-hoc tests for the main effect of condition ($ps > .066$). The *Prime* x *Condition* was not significant $F(3, 175) = 0.43$, $p = .735$, $\eta_p^2 = .007$. Significance was not changed when the model included gender, ethnicity, age, student status, positive and negative affect, or scores on any of the questionnaires described above. These covariates—including age and student status—did not independently correlate with the ratings of the authors in any condition ($|r_s| < .21$; $ps > .1$).

An initial goal of this study was to examine experiential avoidance via the AAQ as a moderator. The model including the AAQ as the only moderator was significant, $F(8, 185) = 3.16$, $p = .002$, $R = 0.35$, $R^2 = .12$. However, only the main effect of dependent variable was significant, $p < .001$, such that the pro-American essay was rated higher than the anti-American essay. No other main effect or interaction was significant in this model. Examining only the anti-American essay resulted in a null model, $F = .29$, $p > .96$.

## Exploratory moderation analyses

**Age.** As above, we conducted regression analysis to probe a potential influence of age. More specifically, we entered essay evaluation as the dependent variable and age, student status, condition, prime, and prime x condition interaction term as the independent variables. The model was significant, $F(9, 205) = 2.76$, $p = .005$, $R = 0.33$, $R^2 = .11$. Only essay type emerged as a significant predictor in the model whereby the pro-American writer was evaluated more positively than the anti-American writer across all conditions, $b = -1.016$, $SE_b = 0.44$, 95% $CI_b$ (-2.03, -0.29), $t = 2.62$, $p = .021$, $\beta = -.68$. No other predictor approached significance in this model ($ps > .09$). Finally, to reflect the regression models above, we also ran an additional analysis with only the anti-American ratings as the dependent variable, and prime, age, and the prime X age interaction as independent variables, but the model was not significant, $F < 0.5$, $p = .83$, $R = 0.19$, $R^2 = 0.03$, with no significant predictors. A similar model with prime, student status and the prime X student-status interaction was also not significant, $F = .72$, $p = .66$, $R = 0.22$, $R^2 = 0.05$, with no significant predictors within the model. Adding age to this latter model, the age X student status, and the age X student status X condition interaction terms did not improve model fit, $F = .68$, $p = .77$, $R = 0.29$, $R^2 = 0.08$.

**Religiosity.** As an exploratory step, we took a closer look at religiosity and self-esteem to reflect the analyses conducted in Studies 1–4. First, we ran a regression model with prime, essay type, religiosity, the interaction terms of religiosity x prime and religiosity X essay type. The model was significant model as before, $F = 3.01$, $p = .002$, $R = 0.35$, $R^2 = 0.12$. Examination of the predictors indicated a trend-level decrease in author ratings as religiosity increased, $b = -.04$, $SE_b = 0.02$, 95% $CI_b$ (-0.08, 0.00), $t = 1.97$, $p = .050$, $\beta = -.289$. This main effect is qualified by a trend-level interaction with essay type (pro- vs. anti-American), $b = -0.04$, $SE_b = 0.02$, 95% $CI_b$ (-0.00, -0.08), $t = 1.96$, $p = .051$, $\beta = .27$. Surprisingly, examining simple slopes of this interaction indicated that, as religiosity increased, ratings of the pro-American essay decreased in positivity while ratings of the anti-American increased.

**Self-esteem.** Regression analyses with prime, essay type, self-esteem, the interaction terms of self-esteem x prime and self-esteem X essay type resulted in a significant model, $F = 2.47$, $p = .011$, $R = 0.31$, $R^2 = 0.10$. Examination of the coefficients again showed the pro-American essay to be rated more positively than the anti-American essay, as would be expected, $b = -0.87$, $SE_b = 0.23$, 95% $CI_b$ (-1.32, -0.42), $t = 3.81$, $p < .001$, $\beta = -.51$. However, no other variables were significant predictors of the author ratings.

## Study 6: Summing up studies 1–5 via meta-analysis

As a final strategy to examine the effects of TMT on worldview defense, we conducted a "mini" meta-analysis across all the studies we reported. We conducted this meta-analysis to facilitate broader purview of these data and address potential concerns about power—a consistent criticism for any failure to replicate [51, 83]. Each correlation coefficient underwent Fisher's $r$-to-$z$ transformation. We used a random-effects model because we had no assumption of an underlying population mean [84] and we calculated between-study variance using restricted maximum likelihood estimation, which is a less biased estimator of variance

compared to other methods [85]. Because the effects we examined contained different metrics (e.g., bail vs. scale), we first computed the meta-analysis using the correlations between MS and each dependent variable in the studies we report—bail for arrested prostitute (Studies 1 and 2), evaluation of anti-American essay (studies 1–5), and fear (Studies 3 and 4)—with a pooled sample of 1255 responses across all conditions. Results of this analysis are presented in Fig 1. We then extracted just the evaluation of the anti-American essays from each study and evaluated them on their own with a pooled sample of 651 responses as this essay was the only variable that was consistent across all five studies and is arguably the clearest (or at least most classic) measure of worldview defense available in these studies. The results of this partial model are presented in Fig 2. We used the "meta" package [86] in RStudio (RStudio Team, Boston, MA). Note that these analyses are exploratory to parsimoniously examine the overall effect of MS in the present set of studies and to probe whether effects may have been lost to Type II error due to insufficient sample size for individual analyses. Analyses of both models yielded a null effect: the overall effect sizes (*r*) for both models were nearly zero.

## Discussion

The present set of studies share a common theme: each was intended to be a replication *and* extension of TMT using prototypical MS primes, yet none were able to successfully replicate a basic MS effect. Across five studies with 1255 unique participants, anti-American or anti-American-values essays were evaluated either comparably or more positively than control conditions. Further, 651 participants who completed a variety of classic MS primes rated the author of an anti-American essay either comparably or more positively than control conditions. It was not just a lack of statistical significance—which clearly is not the only metric for a "successful" study or supported hypothesis—it was that, despite similar methods used across studies, the pattern of effects was inconsistent and/or the effect sizes were negligible. One would expect the MS prime to result in significantly more negative ratings of an anti-American

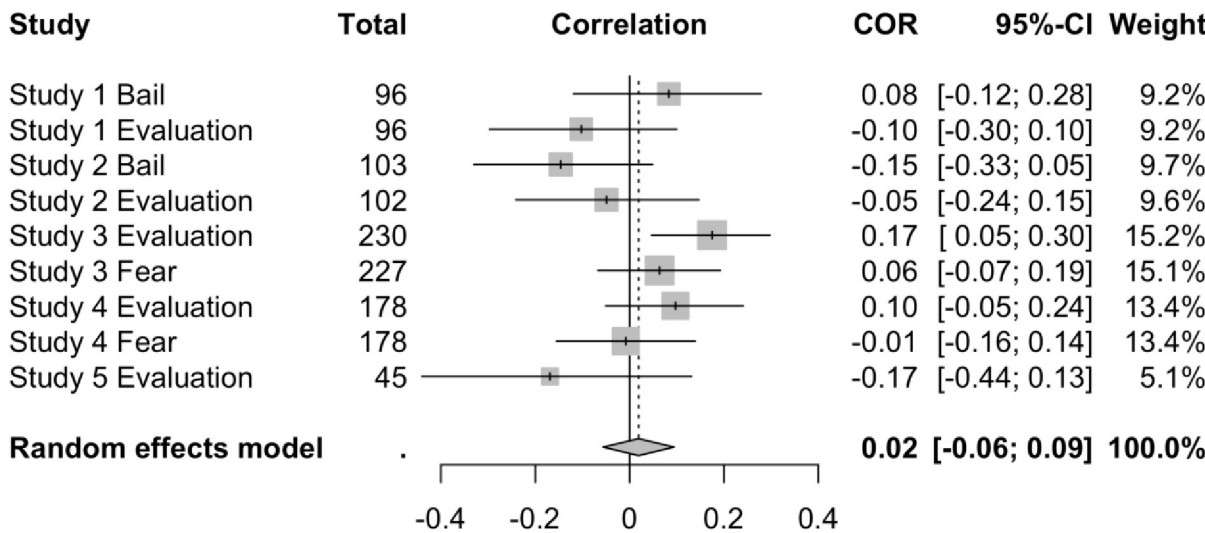

**Fig 1. Forest plot of effect sizes from all studies in this manuscript ($N_{pooled}$ = 1255).** Total = total *n* for each study (different *n*s within each study reflect participants completing one measure but not the other). Evaluation = evaluation of an anti-American essay. For measures of bail and fear, positive correlations reflect higher scores in the MS than in the control condition. For evaluation measures, negative correlations are hypothesis-consistent and reflect greater worldview defense in the MS condition than in the control condition. For the meta-analytic effect, $z$ = 0.50, $p$ = .618, $\tau^2$ = .0053, $I^2$ = 40.80%, 95% $\text{CI}_I^2$ = (< .01%, 72.70%).

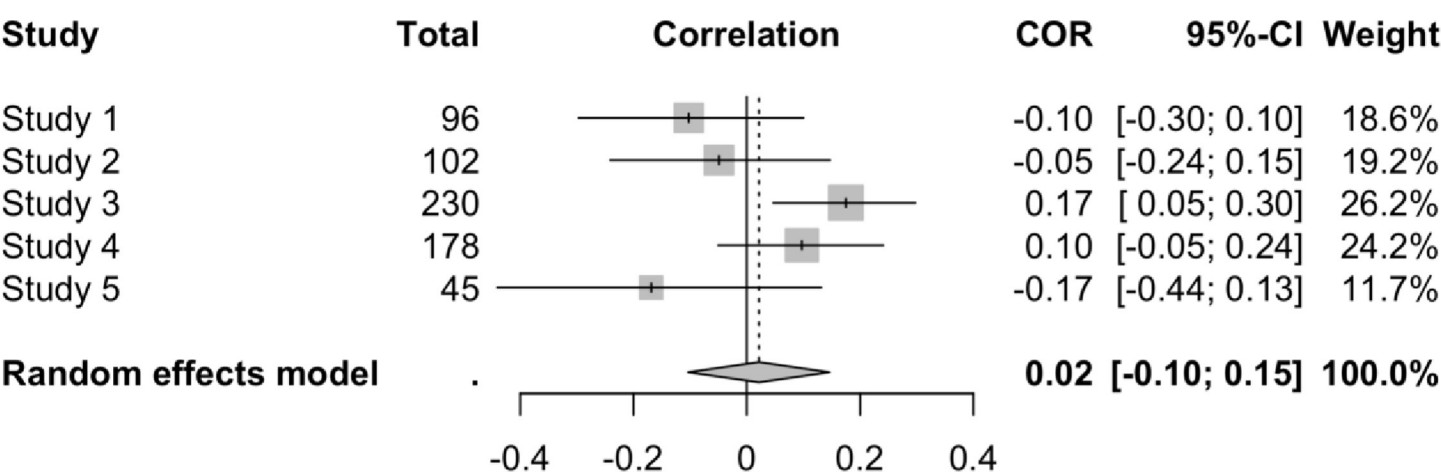

**Fig 2. Forest plot of only effect sizes evaluating an anti-American essay across all studies analyzed in this manuscript ($N_{pooled}$ = 651).** Total = total $n$ for each study. Negative correlations are hypothesis-consistent and reflect negative evaluations of the anti-American essay's author (worldview defense) in the MS condition than in the control condition. For the meta-analytic effect, $z = 0.33$, $p = .738$, $\tau^2 = .0112$, $I^2 = 56.50\%$, 95% CI$_I^2$ = (< .01%, 83.90%).

essay, more positive ratings of a pro-American essay, or demonstrate some degree of greater worldview defense [8], yet we did not observe these here and it is unclear why.

In closely examining why a study failed to replicate a previously established phenomenon, it is important to evaluate the stimuli, method of administration, and sample characteristics [87]. Evaluating these methods closely is critical to rule out an "operational failure," which prevents a valid hypothesis test [88]. In terms of the stimuli used, we took care to reproduce the methods used in studies as closely as possible using publicly available original stimuli (with added conditions as study interest dictated). And all evaluations of worldview defense (essay and author evaluation) were identical to those used in foundational research in this field [5, 27, 82, 89]. Other possible methodological differences are discussed below.

## Method of administration

A key distinction between the present studies and much of the previous TMT literature was our use of MTurk and online administration rather than a more typical lab-based or in-person administration. Some argue that many of the contemporary failures to replicate classic social and personality psychology studies are due to the reduced motivation and engagement by remote participants completing studies asynchronously—on their own time and in the setting of their choice [88]. However, compared to undergraduate research pool respondents, online study participants typically provide comparable or superior data quality and research fidelity [e.g., 90]. Although some recent concerns about MTurk quality have emerged, data support that diverse validity and attention checks, such as those included in the present studies, adequately protect data quality [90–92]. In TMT research, the results of online samples have been mixed. Some online TMT research has resulted in atypical results [91] and failures to replicate [32, 47]. Yet there are numerous other published studies that have been successful in inducing MS using essay primes via online or MTurk administration [93–95]. Online samples have also successfully exhibited MS effects when encountering indirect primes (e.g., ads on a website), questionnaires, or word fragment completion [26, 96–99]. Therefore, online administration, in-and-of-itself, unlikely explains the results observed in the present studies. We also find it unlikely that online administration could explain the null effects observed across *five* different samples, one of which (Study 5) was predominantly in-person, especially as we included measures of seriousness and social desirability. For Study 5, including method of administration to

the linear models in these analyses did not change the pattern of results, and method of administration did not emerge as a significant predictor to author ratings (data not shown). These attention- and engagement-check systems are, of course, not perfect, but have been found to detect most poor response quality in previous work [90, 92]. Given established effects of researcher [30, 35] and experimental style [e.g., 54], it may be that this *specific* method does not work in this context. Indeed, many, but not all, of the recent failures to replicate using this method have used online or crowd sourced samples [e.g., 47, 49, 50]. Together, the narrowest interpretation of our results is that online and/or crowdsourced administration of the prototypical essay prime and author evaluation does not work, though it is not clear why.

The constraints of crowd-sourced data may have contributed to the null or tenuous effects that we observed. In crowd-sourced settings such as MTurk, respondents are often incentivized to respond as attentively, compliantly, and quickly as possible to both open higher-paying opportunities and prevent being rejected from future employment [90, 100]. By being motivated to be attentive and compliant, it is certainly possible that participants are in a "rational" mindset in discussing death, thereby potentially reducing the MS effects [54]. Additionally, it was impossible to control or monitor the delay between prime and essay rating, which, in combination with the motivation for speedy responding, may have thwarted the desired effects [21, 23, 101]. We used similar delay methods (e.g., the administration of the PANAS) as previous work, which should have provided the appropriate delay time [23], and also confirmed that the mood of the participant was not measurably impacted by the induction [cf. 71]. All participants completed the delay tasks, each of which is a well-established and/or prototypical measure [23, 24]. However, we could not measure the duration of the delay nor level of engagement with the delay tasks (or any aspect of these tasks), which is certainly something to consider for future work.

Given the proliferation of MTurk (and similar) methods of data collection, it may also be that users are regularly exposed to the same stimuli or questionnaires [102]. Repeated exposure to stimuli can blunt effect sizes in various experimental paradigms [103]. It is not clear what repeated exposure to the stimuli may do, though some previous work suggests it would enhance the effects of MS rather than blunt it [104]. Unfortunately, we had no means to measure whether participants were naïve to the MS conditions, which is a potentially fruitful avenue for future work. Since researchers are increasingly reliant on crowd-sourced data [105, 106], it is important to consider how online-administered studies may impact results, particularly as related to TMT.

Several findings within these samples were reflective of previous work, albeit inconsistently across samples (see Table 1), making it difficult to suggest that online task administration entirely accounted for the results presented here. For example, self-esteem and religiosity interacted with condition in the expected manner (bolstering worldview defense) in Studies 1–3. However, these interactions were not observed in Studies 4 and 5. Examination of self-esteem and religiosity were not central to the original study designs and, therefore, should be interpreted with caution. Further, Type I error cannot be ruled out for these results, given the sheer number of analyses we conducted in the interest of exhausting any possibility of explaining our (null) main findings. Nevertheless, the main and most critical component—worldview defense—was not observed on its own in any sample. In the following, we discuss other potential explanations for the results in the present study.

## Sample characteristics

**Sample size.** In light of the reproducibility crisis in psychological science , it is crucial to evaluate a study's sample size to ensure the results are not simply due to insufficient power

**Table 1. Overview of five included studies in the present manuscript.**

| Study | Final $N$ | Additional DV[*] | Worldview defense result | Additional pretest moderators/analyses |
|---|---|---|---|---|
| 1 | 203 (148 control; 55 MS) | Bail for prostitutes | Bail: Null[†] | Attitudes toward prostitution, self-esteem[¶], religiosity[¶], endorsement of American values |
| | | | Essay: Null[†] | |
| 2 | 103 (52 control; 51 MS) | Bail for prostitutes | Bail: Null[‡] | Attitudes toward prostitution, self-esteem[‡,a], religiosity[¶,b], political orientation, endorsement of American values, religiosity |
| | | | Essay: Null[‡] | |
| | | | Fear: Null[‡] | |
| 3 | 230 (120 control; 110 MS) | Fear | Essay: significant but opposite of TMT Fear: Null[†] | Self-esteem[c], Political orientation, endorsement of American values, religiosity |
| 4 | 178 (112 control; 52 MS) | Fear | Essay: Null[‡] | Self-esteem, political orientation[e], endorsement of American values[e], religiosity |
| | | | Fear: Null | |
| 5 | 183 (Anti-American: 69 Control; 21 MS; Pro-American: 67 Control; 26 MS) | Pro-American essay | Anti-American Essay: Null[†] | Self-esteem, religiosity[‡], experiential avoidance (see text for additional covariates) |
| | | | Pro-American: Null[‡,d] | |

*Notes*: Please see text for all abbreviations, for analyses of any additional measures, and for more details about group and sample sizes;

[*]All respondents evaluated an anti-American essay;

[†]Means were in TMT-consistent direction, but were not statistically significant;

[‡]Means were in the opposite direction of TMT, but did not approach significance;

[¶]Analysis was in expected direction of TMT and statistically significant;

[a]In bail condition only and only in post-hoc analyses within a three-way interaction;

[b]Only as an interaction term with self-esteem;

[c]Self-esteem was negatively associated with fear, but did not interact with condition;

[d]The MS condition also rated the pro-American essay more negatively than the TV condition (results are complex as a function of control group type; see text);

[e]The interaction of political orientation and endorsement of American values with condition and self-esteem was complex, but broadly TMT-consistent (see text).

[83]. In the meta-analysis by Burke, Martens (19), the average sample size of each included study was 87.3 (SD = 50.8) with a remarkable range of 17 to 343. The average effect size reported by Burke and colleagues was $r = .35$ (corresponding to $d = .75$). In the present studies, the smallest sample for a main effect of prime was in Study 1 ($n = 107$), with 47 in the neutral (watching TV) condition compared to 60 in the MS condition. For study 1, sensitivity analysis indicated this sample could have adequately powered (1-$\beta$ = .80; i.e., 80% power) an effect size $d > .44$; and could adequately power an effect size at $d > .70$ at 95% power. Both effect sizes were still smaller than the pooled effect from Burke and colleagues [19], though larger than the one observed by Schindler and colleagues [30], the latter of which postdated our data collection by several years. For Study 5, which contained the greatest number of conditions and the smallest $n$ per cell, in the anti-American essay, the sample was also adequately powered (1-$\beta$ = .80) to detect smaller effect sizes than those in Burke et al. [19]: all control groups ($n = 69$) vs. mortality ($n = 26$): $d = .71$ or $r = .33$; the TV only ($n = 24$) vs. mortality ($n = 26$): $d = .65$ or $r = .31$. This study was not designed to directly compare those two conditions. Original *a priori* power-analyses for the intended 4 (prime) x 2 (essay type) ANOVA indicated 152 responses would adequately power (1-$\beta$ = .80) a medium effect size ($f > .25$; $\eta_p^2 > .058$). The obtained sample ($n = 179$) exceeded this threshold. Confirming the remaining studies, Study 2 ($n_{MS}$ = 52; $n_{control}$ = 51) could power $d = .55$ ($r = .27$), Study 3 ($n_{MS}$ = 110; $n_{control}$ = 120) could power $d = .37$ ($r = .18$), and Study 4 ($n_{MS}$ = 66; $n_{control}$ = 112) could power $d = .44$ ($r = .22$) with 1-$\beta$ = .80 for each. Again, all samples could power effect sizes smaller than Burke and colleagues' pooled effect size [19], though, admittedly, none were powered sufficiently for Schindler and colleagues' [30] pooled effect size. Yet, combining the effects into two small meta-analyses with 1255 datapoints across all conditions, and 651 specifically anti-American essay

evaluations, *still* resulted in null effects and negligible effect sizes. In short, the null effects observed here are unlikely a result of insufficient power.

**Race, ethnicity, nationality.** Race, ethnicity, and nationality have been shown to moderate the impact of MS paradigms [19]. The use of online recruitment methods allowed us to recruit a more diverse and potentially more culturally representative sample than most previous studies, which may result in reduced, or even reversed MS effects. More specifically, effects of MS are attenuated and/or reversed when samples are ethnic minorities and/or are from non-WEIRD cultures [12, 19, 35, 37]. In the case of the present study, we limited analyses to US residents. Unfortunately, we did not collect racial or ethnic identity status for studies 1–4 as, again, these variables were not relevant to the original intention of the studies. Controlling for ethnic identity and student status in Study 5 did not alter significance or the direction of the results. It is impossible to entirely rule out ethno-racial influences as explaining the results in the present study. It is important to be mindful that ethnicity and race do not influence worldview defense on their own; it is problematic to rely solely on these identities as a variable [107, 108]. Future work would do well to monitor sociodemographic variables while also being mindful to explain *why* such differences may emerge.

*Age.* Age is negatively associated with MS effects and older adults exhibit the opposite impact of MS inductions by expressing more egalitarian views [28, 29]. The relationship between age and the effects of the task in the present study were tenuous and inconsistent. Significant findings emerged in Study 4 and as a post-hoc finding despite an insignificant linear model in Study 1. In both, the anti-American essay writers were rated worse as age increased, but this effect did not interact with condition. The association with age is likely accounted for by generational and age-related trends in patriotism [109, 110; discussed further below]. Nevertheless, MS had no demonstrable influence over this pattern in the present study.

The mean age for participants across all studies in the present paper was 34.8 years old (SD = 12.5), which is significantly older than the average of studies reported in Burke, Martens (19), M = 22.2, SD = 4.7; $t$ (280) = 5.70, $p < .001$, $d = 1.33$, and Burke, Kosloff (56), M = 22.9 years, SD = 4.30; $t$ (34) = 4.19, $p = .002$, $d = 1.27$. No other meta-analysis presented the ages of the samples included in their studies. Only Burke, Martens (19) analyzed age within their meta-analysis and did not find it to significantly moderate the effects of mortality salience. Each sample in the present study was older than is typically observed in the TMT literature ($\approx$ 35 years). However, the majority of each sample was under 60 years of age, below the threshold where MS begin to reverse [111]. We cannot unequivocally state the age range of the samples in the present studies did not impact these results, though it seems unlikely. There is a surprising dearth of research in the role of age in TMT, and much more work is needed to better determine whether MS effects are limited to youth and young adults.

**College students.** Interestingly, Burke, Martens (19) found that undergraduates exhibited greater effects of MS than non-students, regardless of age. However, other failures to replicate also included young and/or undergraduate samples [47–50]. In studies 1–4, we did not analyze whether participants were undergraduates, which is a limitation of the present analyses. Notably, the sample in Study 5 was majority (85%) undergraduate students who completed the task in person. However, the *control* condition in Study 5 showed greater pro-national biases, even when controlling for several well-established moderators, including student status, ethnicity, and self-esteem. Therefore, although there may have been a preponderance of non-undergraduates and ethnic minority respondents in the samples of Studies 1–4 this was not the case for Study 5 and does not explain our failure to replicate in that sample.

In summary, the present findings failed to replicate basic MS inductions despite inclusion of five adequately powered samples exclusively from the United States and using well-established methods. Though there were certain methodological areas highlighted above that could

result in *some* reduction in effect, we believe these do not amount to an "operational failure" across five studies [88]. These null findings add to a growing list of recent TMT studies that failed to replicate basic MS effects in reasonable, or even ideal, conditions [47–50]. Methodologically, although we cannot state this unequivocally, we do not believe the online administration of the tasks, per se, resulted in the null results observed here as others have done so successfully. However, the role of the demands of MTurk and crowdsourced data also cannot be ruled out, and merits further exploration (yet, still, does not explain the null results in Study 5). There are several sample characteristics for which we cannot rule out as having influenced the results, including: (a) the older age range of the present samples; (b) the ethnic composition of the samples; (c) the (undergraduate) student status of the samples.

## General discussion

In placing these results in the context of a growing number of failures to replicate, we suggest there are two overarching frameworks in which these findings can be understood. The first is that thoughts of death, at least as induced with essay primes, was never as "terrifying" as suggested. It is not the argument by these authors that nearly four decades of work related to TMT is suddenly moot, or one big "false positive" as Klein and colleagues [49] assert. We do believe that thoughts of one's mortality can transiently influence behaviors and attitudes that, broadly, reduce egalitarian beliefs and increase in-group biases. As well, we do not refute the probability that many cultural structures were designed to reduce anxiety related to mortality awareness. We realize that MS and TMT exist as a part-whole relationship, whereby MS is one (of many) means to measure the components of TMT [1]. However, previous work has found MS primes to be most effective in large samples of late adolescents and young adults who are White, North American undergraduates with low or fragile self-esteem, who complete inductions with precise timing during in-person administration [19, 21, 23, 35]. Although this slice of the population can disproportionately inflict harm when faced with perceived threat [112–115], if MS effects using essay primes are only effective in these samples under those conditions, the explanatory power of this method may be more limited than is often presented. It is certainly possible that, for most of the rest of the world, explicitly thinking and writing about death was, and is, *not so terrifying after all*. We argue that the measurement of TMT needs updating and re-evaluation to explain how components of TMT, especially MS, may influence the general (global) population. And, as mentioned, the validity and utility of these methods needs to be confirmed for use in crowd-sourced data collection as this method is increasingly used in psychological science [105, 106].

A second non-exclusive possibility is that the classic MS inductions used in these studies may simply not work *anymore*, or what Baumeister and colleagues [88] call a *boundary condition* In fact, a recent meta-analysis found a small, but significant decline in effect sizes as a function of date of publication [30]. The MS induction methods used in the present study were developed nearly 40 years ago [e.g., 89]. Since that time, key contextual factors have shifted that may make explicit MS inductions a less unusual or terrifying experience regardless of administrative style. The composition of the global population, especially in the United States, has rapidly changed and continues to do so: it is increasingly older, educated, literate, racially diverse, tolerant, and urbanized [116–120], while also increasingly diverse in religious beliefs, including adherence to no religion [121]. Each of these factors has been shown to attenuate the experimental effects of MS [14, 19, 35, 37, 101]. There is emerging evidence that death is no longer the taboo topic it once was, and the US population is increasingly comfortable discussing mortality and death [122–126]. Some suggest a key mechanism behind this shift in attitude is that self-esteem and symbolic immortality are faster and easier to achieve than in

the past, ostensibly due to the proliferation of technology and social media that provide instantaneous "digital immortality" [7, 127].

The realities of death may not be as abstract for younger "Millennials" and "Generation Z" who largely comprised the samples in this study. These generations entered, or are entering, adulthood post-9/11, or were born after the attacks. After years of decline, rates of adolescent and young adult mortality have increased in the past decade, led by suicides, drug overdoses, and shootings, much of which receives abundant (social) media coverage [128, 129]. There is a new and growing portion of popular media that focuses on death, much of which is intended for children and adolescents [130–132]. Although the data collected here predate the global COVID-19 pandemic by several years, this epoch, its sequelae, and subsequent upheaval are impacting views of mortality in ways we are only beginning to ascertain [133–135] . Taken together, it may be that writing about death is no longer the existential fear- or anxiety-inducing task it may have been in the past: it is not so terrifying *anymore*.

A final consideration is the dependent variable of choice in the standard MS induction tasks—national pride or adherence to (conservative) American values—may no longer be a salient measures of worldview defense. In fact, a recent meta-analysis found that, when accounting for researcher effects and publication bias, worldview defense, specifically, was not impacted by MS primes across studies [30]. The authors further highlight that researchers infrequently confirm that the worldview being assessed within their work is salient for their participants, or, put another way, is one that participants would *ever* defend with or without MS [30]. In the case of the present studies, mean endorsement of American values did not moderate the present results. The mean score on the measure was 4.03 (SD = 1.46) across all samples, indicating 57% endorsement of American values. Though not an overwhelming endorsement, it is still sizable. Therefore, it is unclear if these samples, overall, would defend these worldviews outside of the MS primes. Nevertheless, the precise aspect of American values may not be a worldview the participants in this sample would defend. Namely, the pro- and anti-American essays discuss economic opportunities and socioeconomic mobility—the so-called "American dream." There is growing pessimism about socioeconomic mobility in the United States [136, 137]. Though there is evidence that contemporary youth and young adults still, on the whole, believe in the "American dream," that belief remains almost exclusively held by already-wealthy White men [138], or college graduates who have more upward mobility than other demographics [137, 139].

Pro-national and patriotic attitudes of Americans, especially younger Americans, have declined over time [109, 110]. Further, growing numbers of Americans, especially younger Americans, report greater distrust and criticism of the United States than in the past several decades [140, 141]. Contemporary younger adults' political views are considered more "liberal" than prior generations at the same age [142], including attitudes toward sex workers as assessed in Studies 1 and 2 [143, 144]. Thus, it may no longer be that rating the authors of an anti-American essay (which are really pro- or anti-American *dream* essays) and/or setting bail for prostitutes are sufficient to index the worldview the average participant will be inclined to defend in-lab or otherwise. Indeed, ample previous work has established that liberal people defend liberal worldviews during existential threat [9], which may be an increasingly common occurrence if generational political trends hold as they seem to be. In the case of the present results, it may be that that we unwittingly activated politically liberal worldview defenses that we were not prepared to measure. It is important for future work to consider these shifting baseline attitudes when designing studies, perhaps by including egalitarian worldviews to defend.

## Limitations and conclusion

None of these studies were *intended* to be "failures to replicate," which we argue is both a limitation and a strength. As a limitation, we did not intend to challenge or support these methods and, therefore, did not gather as much sociodemographic information as would be ideal. The strengths of these studies lie in reduced bias of the authors (i.e., putting in variables that pull results toward, or away from, statistical significance) than if we had pursued a pre-registered report—which is not a panacea for bad methodology and dishonesty—with strict methodological parameters. Some view even slight deviations from methods in pre-registered replications as misleading [145]. The combined examination of these five studies could be considered "ecologically valid" in that our *a priori* goals likely reflect those of the broader research community than those trying to support or refute a theory. Moreover, we suggest our results may look like a lot of studies that ended up in the so-called "file drawer" [57]. Going forward, it is critical for researchers, no matter the intention to replicate, to closely monitor their sample composition and methods to evaluate how thoughts of death influence attitudes, beliefs, emotions, and behavior. Further, we support and encourage future researchers to engage in transparency and provide open data—as we have here—or other methods to increase trust in their findings. It is important to submit failures to replicate, unintentional or otherwise, for the broader field to and scrutinize. As Spellman (58) eloquently states: ". . .science proceeds not only by the accretion of new facts but also by the weeding out of what was once falsely believed" (p. 59).

## Acknowledgments

The authors would like to thank Thomas F. Price for his help on these studies and on earlier drafts of the manuscript.

## Author Contributions

**Conceptualization:** Stanislav Treger, C. Alix Timko.

**Data curation:** Stanislav Treger, Erik M. Benau, C. Alix Timko.

**Formal analysis:** Stanislav Treger, Erik M. Benau.

**Investigation:** Stanislav Treger, C. Alix Timko.

**Methodology:** Stanislav Treger.

**Project administration:** Stanislav Treger.

**Supervision:** C. Alix Timko.

**Visualization:** Stanislav Treger.

**Writing – original draft:** Stanislav Treger, Erik M. Benau.

**Writing – review & editing:** Stanislav Treger, Erik M. Benau, C. Alix Timko.

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
