## [Decision Letter · Decision Letter 0]

15 Feb 2023

PONE-D-22-26798Not so terrifying after all? A set of failed replications of the mortality salience effects of Terror Management TheoryPLOS ONE

Dear Dr. Benau,

Thank you for submitting your manuscript to PLOS ONE. After careful consideration, we feel that it has merit but does not fully meet PLOS ONE’s publication criteria as it currently stands. Therefore, we invite you to submit a revised version of the manuscript that addresses the points raised during the review process.

We look forward to receiving your revised manuscript.

Kind regards,

Boshra Ismael Ahmed Arnout

Academic Editor

PLOS ONE

Additional Editor Comments:

Dear Author

The paper PONE-D-22-26798 has been reviewed by experts in the field which consider that the paper can publish after minor revision. For your guidance, reviewer's comments are appended below.

We wish you a meaningful day.

Yours Sincerely

Reviewers' comments:

Reviewer's Responses to Questions

**Comments to the Author**

1. Is the manuscript technically sound, and do the data support the conclusions?

Reviewer #1: Yes

Reviewer #2: Yes

2. Has the statistical analysis been performed appropriately and rigorously? 

Reviewer #1: Yes

Reviewer #2: Yes

3. Have the authors made all data underlying the findings in their manuscript fully available?

Reviewer #1: Yes

Reviewer #2: Yes

4. Is the manuscript presented in an intelligible fashion and written in standard English?

Reviewer #1: Yes

Reviewer #2: Yes

5. Review Comments to the Author

Reviewer #1: Across five experiments with various manipulations and independent sets of participants, the present study demonstrated a robust case of failure of replicating the mortality salience (MS) effect. I recommend the publication of this work after the authors address the following minor concerns.

1. Is there any way to make sure participants took the filler task (i.e., the one between the MS induction phase and target dependent measures) seriously? In other words, were they indeed fully occupied during the filler task?

2. As noted by the authors, some of the findings did reflect TMT’s hypothesis that higher self-esteem decreased worldview defense following MS and that religiosity may bolster one’s worldviews to buffer negative effects of MS. This should be discussed with details in the (General) Discussion section.

3. I suggest the authors to take a look on the following work, which also showed that the mortality salience effect seems to depend on cultural orientation.

Zeng, T., & Tse, C.-S. (2020). Does the mortality salience effect on worldview defence depend on the cultural orientation of Chinese people? International Journal of Psychology, 55, 291-304.

Reviewer #2: Comments to the manuscript “Not so terrifying after all? A set of failed replications of the mortality salience effects of Terror Management Theory" (Manuscript ID: PONE-D-22-26798):

Please discuss more critically the discussion on the problems of replication of the basic findings of TMT and especially the problem of false positive results/p hacking in this field.

Please also discuss in detail the meta-analysis of Schindler et al. (2022) that critically pointed to the problems of publication bias in TMT research:

Schindler, S., Hilgard, J., Fritsche, I., Burke, B., & Pfattheicher, S. (2022). Do salient social norms moderate mortality salience effects? A (challenging) meta-analysis of terror management studies. Personality and Social Psychology Review. Advance online publication. https://doi.org/10.1177/10888683221107267

Please explain in more details the main hypothesis of the 5 reported studies.

Please reorganize the result part more like a “meta-study”. One method section describing the parallel and specific methods of all experiments. Moreover the results of all experiments could be presented in one result part using Tables to describe the samples and results.

Please report sensitivity analysis to specify what effect (effect size) could be detected in the experiments.

In sum, the results are important given that the replicability of basic findings of TMT is clearly in question.

6. PLOS authors have the option to publish the peer review history of their article (what does this mean?). If published, this will include your full peer review and any attached files.

Reviewer #1: No

Reviewer #2: No

---

## [Author Response · Author response to Decision Letter 0]

30 Mar 2023

We have reformatted accordingly. 

All raw data, survey materials, STATA 13 (for studies 1-4; StataCorp, College Station Texas), RStudio (mini meta-analysis; RStudio Team, Boston, MA) code, and SPSS (IBM inc., Armonk, NY) syntax (Study 5) used to report the analyses are available at the study’s OSF page at: https://osf.io/qda5b/?view_only=8618abc1cf6e4ddc811a9d92dd46783f. This statement also appears in the manuscript (see page 12). We have confirmed that this link is working. Please let us know if we need to provide anything further.

This is only located in the methods section. We also relabeled “Overview of Studies “to “Methods” to make that clearer. We adjusted the headings in this section to reflect the change.

So far as we know, no paper we are citing has been retracted as of this writing. If the editor is aware of one, please let us know and we will find an alternative or justify its use.

5. Reviewer #1: Across five experiments with various manipulations and independent sets of participants, the present study demonstrated a robust case of failure of replicating the mortality salience (MS) effect. I recommend the publication of this work after the authors address the following minor concerns.

Thank you for these comments and for taking the time to provide helpful feedback.

1. Is there any way to make sure participants took the filler task (i.e., the one between the MS induction phase and target dependent measures) seriously? In other words, were they indeed fully occupied during the filler task?

Though we cannot assess the duration of the delay or the seriousness with which our participants engaged in these. We have made that more specific in the methods and discussion sections.

Pg. 9: Additionally, all participants included in the present analyses provided complete data, including responses on the filler task(s) used to cause the requisite delay and distraction between prime and dependent variables [23, 24].

Pg. 47: All participants completed the delay tasks, each of which is a well-established and/or prototypical measure [23, 24]. However, we could not measure the duration of the delay nor level of engagement with the delay tasks (or any aspect of these tasks), which is certainly something to consider for future work.

2. As noted by the authors, some of the findings did reflect TMT’s hypothesis that higher self-esteem decreased worldview defense following MS and that religiosity may bolster one’s worldviews to buffer negative effects of MS. This should be discussed with details in the (General) Discussion section.

The author does raise an important point here, and one that provides further support that method of administration alone cannot explain our null findings. Namely, if we observed some elements of TMT-consistent effects in these samples, then it becomes difficult to dismiss our findings due to administration problems. We have added a paragraph in the discussion section expounding on this point. Additionally, so that all of our analyses are parallel to emphasize this effect, we conducted additional, focused regression models in Study 5 to fully rule out or rule in religiosity and self-esteem as interacting with the variables (religiosity resulted in the opposite effect as would be expected, and self-esteem resulted in no effect). 

Pg. 40-41: As an exploratory step, we took a closer look at religiosity and self-esteem to reflect the analyses conducted in Studies 1-4. First, we ran a regression model with prime, essay type, religiosity, the interaction terms of religiosity x prime and religiosity X essay type. The model was significant model as before, F = 3.01, p = .002, R = 0.35, R2 = 0.12. Examination of the predictors indicated a trend-level decrease in author ratings as religiosity increased, b = -.04, SEb = 0.02, 95% CIb (-0.08, 0.00), t = 1.97, p = .050, β = -.289. This main effect is qualified by a trend-level interaction with essay type (pro- vs. anti-American), b = -0.04, SEb = 0.02, 95% CIb (-0.00, -0.08), t = 1.96, p = .051, β = .27. Surprisingly, examining simple slopes of this interaction indicated that, as religiosity increased, ratings of the pro-American essay decreased in positivity while ratings of the anti-American increased. 

Pg. 47-48: Several findings within these samples were reflective of previous work, albeit inconsistently across samples (see Table 1), making it difficult to suggest that online task administration entirely accounted for the results presented here. For example, self-esteem and religiosity interacted with condition in the expected manner (bolstering worldview defense) in Studies 1–3. However, these interactions were not observed in Studies 4 and 5. Examination of self-esteem and religiosity were not central to the original study designs and, therefore, should be interpreted with caution. Further, Type I error cannot be ruled out for these results, given the sheer number of analyses we conducted in the interest of exhausting any possibility of explaining our (null) main findings. Nevertheless, the main and most critical component—worldview defense—was not observed on its own in any sample. 

3. I suggest the authors to take a look on the following work, which also showed that the mortality salience effect seems to depend on cultural orientation.

Zeng, T., & Tse, C.-S. (2020). Does the mortality salience effect on worldview defence depend on the cultural orientation of Chinese people? International Journal of Psychology, 55, 291-304.

We thank the reviewer for drawing our attention to this informative work. We have added reference to it in the introduction to emphasize that it may be problematic to look at reductions in MS effects by nationality as a monolith, and instead, there are nuanced patterns of MS to be considered.

Pg. 6. However, cultural orientation and relational (vs. personal) self-esteem have emerged as important moderators of MS effects [34], suggesting that “Western” vs. “non-Western” may not be sufficient to understand cultural influences in TMT; and classifying studies that way may be problematic and essentialist. 

Reviewer #2: Comments to the manuscript “Not so terrifying after all? A set of failed replications of the mortality salience effects of Terror Management Theory" (Manuscript ID: PONE-D-22-26798):

Please discuss more critically the discussion on the problems of replication of the basic findings of TMT and especially the problem of false positive results/p hacking in this field.

Though we agree with the reviewer that addressing some of these concerns is likely necessary to provide context for this paper, we hesitate to expand much past acknowledgment of these problems. We believe it is beyond the scope of the present paper to thoroughly elaborate on those concerns. Our goal was to be as transparent as possible regarding the current data. We have added a line to the introduction and conclusions section highlighting this point.

Pg. 7: Others have accused previous researchers of cherry-picking data, presenting false-positives, and otherwise engaging in misleading or less-than-honest behavior (intentionally or otherwise) in studies of TMT [30, 49-51]. However, lapses in rigor and difficulty with replication are far from unique to TMT-related work [52, 53].

Pg. 55-56: Further, we support and encourage future researchers to engage in transparency and provide open data—as we have here—or other methods to increase trust in their findings..

Please also discuss in detail the meta-analysis of Schindler et al. (2022) that critically pointed to the problems of publication bias in TMT research:

Schindler, S., Hilgard, J., Fritsche, I., Burke, B., & Pfattheicher, S. (2022). Do salient social norms moderate mortality salience effects? A (challenging) meta-analysis of terror management studies. Personality and Social Psychology Review. Advance online publication.https://doi.org/10.1177/10888683221107267

We thank the reviewer for drawing our attention to this very informative reference. We now cite it throughout the manuscript and highlight some of the main findings in the introduction and discussion section. The below quotes are the more crucial contributions to the present meta-analysis, though this paper is referred to throughout (for brevity, we only provide these exemplar quotes in this response here). If there are other aspects of the study the reviewer believes we should include, we are happy to consider that in a further revision.

Pg. 7: Although meta-analyses indicate a fairly robust effect of TMT on a host of dependent variables [19], Schindler and colleagues [30] found that the prototypical dependent variable—worldview defense— is not robustly found across studies when accounting for publication bias, control conditions, researcher effects (namely, researcher degrees of freedom), and other key components [30]. In other words, these “conceptual replications,” among others, were inconsistently successful, especially in more recent studies [30, 50].

Pg. 52-53: In fact, a recent meta-analysis found that, when accounting for researcher effects and publication bias, worldview defense, specifically, was not impacted by MS primes across studies [30]. The authors further highlight that researchers infrequently confirm that the worldview being assessed within their work is salient for their participants, or, put another way, is one that participants would ever defend with or without MS [30].

Please explain in more details the main hypothesis of the 5 reported studies.

We now provide a brief overview of the hypotheses for the studies in the “purpose of the studies” section within the methods (pages 10-11). We believe this has added clarity to the manuscript and thank the reviewer for the suggestion.

Please reorganize the result part more like a “meta-study”. One method section describing the parallel and specific methods of all experiments. Moreover the results of all experiments could be presented in one result part using Tables to describe the samples and results.

We thank the reviewer for this comment. We have now added a table with brief synopsis of each study, which help the reader understand the overview of the studies. However, we are unsure what the reviewer means by their suggestion to reorganize like a “meta-study.” We present all shared methods at the beginning of the methods section and any specific variations that occurred are described within each study. We also include a meta-analysis in Study 6. Together, it is our understanding that our paper is already organized like a meta-study. If there is something we are misunderstanding, we are happy to address the concern in a subsequent revision.

Please report sensitivity analysis to specify what effect (effect size) could be detected in the experiments. 

We have now added sensitivity analyses for all studies in the discussion section. These were previously omitted these for brevity, but we agree their inclusion adds transparency. 

Pg. 48-49: “Confirming the remaining studies, Study 2 (nMS = 52; ncontrol = 51) could power d = .55 (r = .27), Study 3 (nMS = 110; ncontrol = 120) could power d = .37 (r = .18), and Study 4 (nMS = 66; ncontrol = 112) could power d = .44 (r = .22) with 1-β = .80 for each. Again, all samples could power effect sizes smaller than Burke and colleagues’ pooled effect size [19], though, admittedly, none were powered sufficiently for Schindler and colleagues’ [30] pooled effect size..” 

In sum, the results are important given that the replicability of basic findings of TMT is clearly in question.

Again, thank you for taking the time to provide thorough and helpful feedback.

---

## [Decision Letter · Decision Letter 1]

19 Apr 2023

Not so terrifying after all? A set of failed replications of the mortality salience effects of Terror Management Theory

PONE-D-22-26798R1

Dear Dr. Benau,

We’re pleased to inform you that your manuscript has been judged scientifically suitable for publication and will be formally accepted for publication once it meets all outstanding technical requirements.

Kind regards,

Boshra Ismael Ahmed Arnout

Academic Editor

PLOS ONE

Additional Editor Comments (optional):

Dear Author

The paper PONE-D-22-26798R1 has been reviewed by experts in the field who consider that the revised paper can publish.

We wish you a meaningful day.

Yours Sincerely

Reviewers' comments:

Reviewer's Responses to Questions

**Comments to the Author**

1. If the authors have adequately addressed your comments raised in a previous round of review and you feel that this manuscript is now acceptable for publication, you may indicate that here to bypass the “Comments to the Author” section, enter your conflict of interest statement in the “Confidential to Editor” section, and submit your "Accept" recommendation.

Reviewer #1: All comments have been addressed

2. Is the manuscript technically sound, and do the data support the conclusions?

Reviewer #1: Yes

3. Has the statistical analysis been performed appropriately and rigorously? 

Reviewer #1: Yes

4. Have the authors made all data underlying the findings in their manuscript fully available?

Reviewer #1: (No Response)

5. Is the manuscript presented in an intelligible fashion and written in standard English?

Reviewer #1: Yes

6. Review Comments to the Author

Reviewer #1: I was Reviewer 1 in the last round of review. The authors responded to my comments and addressed the concerns that I brought up in my review quite satisfactorily and I recommend the publication of this work.

7. PLOS authors have the option to publish the peer review history of their article (what does this mean?). If published, this will include your full peer review and any attached files.

Reviewer #1: No

---

## [Editor Report · Acceptance letter]

28 Apr 2023

PONE-D-22-26798R1 

Not so terrifying after all? A set of failed replications of the mortality salience effects of Terror Management Theory 

Dear Dr. Benau:

I'm pleased to inform you that your manuscript has been deemed suitable for publication in PLOS ONE. Congratulations! Your manuscript is now with our production department. 

Kind regards, 

on behalf of

Professor Boshra Ismael Ahmed Arnout 

Academic Editor

PLOS ONE